

# Identifying Criegee intermediates as potential oxidants in the troposphere

Anna Novelli[1,2], Korbinian Hens[1], Cheryl Tatum Ernest[1,3], Monica Martinez[1], Anke C. Nölscher[1,4], Vinayak Sinha[5], Pauli Paasonen[6], Tuukka Petäjä[6], Mikko Sipilä[6], Thomas Elste[7], Christian Plass-Dülmer[7], Gavin J. Phillips[1,8], Dagmar Kubistin[1,7,9], Jonathan Williams[1], Luc Vereecken[1,2], Jos Lelieveld[1] and Hartwig Harder[1]

[1] {Atmospheric Chemistry Department, Max Planck Institute for Chemistry, 55128 Mainz, Germany}

[2] Now at: {Institute of Energy and Climate Research, IEK-8: Troposphere, Forschungszentrum Jülich GmbH, 52428 Jülich, Germany}

[3] Now at: {Department of Neurology University Medical Center of the Johannes Gutenberg University Mainz, 55131 Mainz}

[4] Now at: {Division of Geological and Planetary Sciences, California Institute of Technology, Pasadena, USA}

[5] {Department of Earth and Environmental Sciences, Indian Institute of Science Education and Research Mohali, Sector 81 S.A.S. Nagar, Manauli PO, Mohali 140 306, Punjab, India}

[6] {Department of Physics., P.O. Box 64. 00014 University of Helsinki, Finland}

[7] {German Weather Service, Meteorological Observatory Hohenpeissenberg (MOHp), Albin-Schwaiger-Weg 10, 83282 Hohenpeissenberg, Germany}

[8]{Department of Natural Sciences, University of Chester, Thornton Science Park, Chester, CH2 4NU, UK}

[9] {University of Wollongong, Wollongong, Australia}



Correspondence to: H. Harder (hartwig.harder@mpic.de)
**Abstract**
We analysed the extensive dataset from the HUMPPA-COPEC 2010 and the HOPE 2012
field campaigns in the boreal forest and rural environments of Finland and Germany,
respectively, and estimated the abundance of stabilised Criegee intermediates (SCI) in the
lower troposphere. Based on laboratory tests, we propose that the background OH signal
observed in our IPI-LIF-FAGE instrument during the afore-mentioned campaigns is caused at
least partially by SCI. This hypothesis is based on observed correlations with temperature and
with concentrations of unsaturated volatile organic compounds and ozone. The background
OH concentration also complements the previously underestimated production rate of sulfuric
acid and is consistent with its scavenging through the addition of sulphur dioxide. A central
estimate of the SCI concentration of $\sim 5 \times 10^4$ molecules cm$^{-3}$ (with an order of magnitude
uncertainty) is calculated for the two environments. This implies a very low ambient
concentration of SCI, though, over the boreal forest, significant for the conversion of $SO_2$
into $H_2SO_4$. The large uncertainties in these calculations, owing to the many unknowns in the
chemistry of Criegee intermediates, emphasise the need to better understand these processes
and their potential effect on the self-cleaning capacity of the atmosphere.
**1    Introduction**
Criegee intermediates (CI), or carbonyl oxides, are formed during the ozonolysis of
unsaturated organic compounds (Criegee, 1975; Johnson and Marston, 2008; Donahue et al.,
2011): in the gas phase ozone attaches to a double bond forming a primary ozonide (POZ)
that quickly decomposes forming a Criegee intermediate and a carbonyl compound. The CI



can exist as thermally stabilised CI (SCI) or as chemically activated CI (Kroll et al., 2001;
Drozd et al., 2011), where the chemically activated CI have high energy content and in the
atmosphere either undergo unimolecular decomposition, or are stabilised by collisional
energy loss forming SCI.
For many decades the chemistry of Criegee intermediates was investigated both with
theoretical and indirect experimental studies as reviewed in detail by Johnson and Marston
(2008), Vereecken and Francisco (2012), and Vereecken et al. (2015). During the last few
years, numerous experimental studies specifically on stabilised Criegee intermediates have
been performed following their first detection by Welz et al. (2012). Many laboratories have
now detected SCI with various techniques (Berndt et al., 2012;Mauldin III et al.,
2012;Ouyang et al., 2013;Taatjes et al., 2013;Ahrens et al., 2014;Buras et al., 2014;Liu et al.,
2014a;Sheps et al., 2014;Novelli et al., 2014b;Stone et al., 2014;Chhantyal-Pun et al.,
2015;Lee, 2015;Newland et al., 2015a;Fang et al., 2016a;Smith et al., 2016) and have
confirmed that they are very reactive towards many atmospheric trace gases. Currently, the
most studied Criegee intermediates are formaldehyde oxide, $CH_2OO$, acetaldehyde oxide,
$CH_3CHOO$ (*syn* and *anti, i.e.* with the outer oxygen pointing towards or away from an alkyl
group, respectively) and acetone oxide, $(CH_3)_2COO$.

| Formaldehyde oxide | Syn-acetaldehyde oxide | Anti-Acetaldehyde oxide | Acetone oxide |

The importance of stabilised Criegee intermediates as oxidants in the atmosphere depends on
the rate coefficient of their reaction with water vapour as the latter is ubiquitously present in



relatively high concentrations in the boundary layer (between $10^{16}$ to $10^{17}$ molecules cm$^{-3}$).
The rate of this reaction strongly depends on the CI conformation (Aplincourt and Ruiz-
López, 2000;Tobias and Ziemann, 2001;Ryzhkov and Ariya, 2003;Kuwata et al.,
2010;Anglada et al., 2011;Anglada and Sole, 2016;Chen et al., 2016;Lin et al., 2016;Long et
al., 2016) and until now the rate coefficient has been measured for *anti*-CH$_3$CHOO (Taatjes
et al., 2013;Sheps et al., 2014) while a lower limit has been determined for CH$_2$OO (Stone et
al., 2014), *syn*-CH$_3$CHOO (Taatjes et al., 2013;Sheps et al., 2014) and (CH$_3$)$_2$COO (Huang et
al., 2015;Newland et al., 2015b). The uncertainties in these rate coefficients make it difficult
to estimate the importance of Criegee intermediates and the impact they may have as oxidants
in the atmosphere. Additionally, recent studies (Berndt et al., 2014b;Chao et al., 2015;Lewis
et al., 2015;Smith et al., 2015;Lin et al., 2016) showed that the reaction between CH$_2$OO and
water dimers (present in the ppmv range in the atmosphere (Shillings et al., 2011)) is faster
than the reaction with water vapor, in agreement with the several theoretical studies
(Ryzhkov and Ariya, 2004;Chen et al., 2016;Lin et al., 2016)which indicate the reaction with
water dimers to be between 400 and 35,000 times faster than the reaction with water vapor
depending on the conformers. Another important reaction of SCI that depends on the SCI
conformation is their unimolecular decomposition. The decomposition rate and product
formed depend on the SCI conformer structure. *Anti*-SCI are likely to isomerise via the ester
channel forming an ester or an acid as final product while *syn*-SCI will form a vinyl
hydroperoxide (VHP) which promptly decomposes forming hydroxyl radicals (OH) and a
vinoxy radical (Paulson et al., 1999;Johnson and Marston, 2008;Drozd and Donahue,
2011;Vereecken and Francisco, 2012;Kidwell et al., 2016). Larger and more complex
conformers such as hetero-substituted or cyclic structures are subject to additional
unimolecular rearrangements (Vereecken and Francisco, 2012). On the unimolecular





decomposition rates and products few experimental data are available (Horie et al.,
1997;Horie et al., 1999;Fenske et al., 2000a;Novelli et al., 2014b;Kidwell et al., 2016;Fang et
al., 2016a;Smith et al., 2016), but more is available from theoretical studies explicitly
focusing on the path followed by different conformers (Anglada et al., 1996;Aplincourt and
Ruiz-López, 2000;Kroll et al., 2001;Zhang and Zhang, 2002;Nguyen et al., 2009b;Kuwata et
al., 2010).
Most of the experimental and theoretical information described above refers to the smaller
conformers. These compounds are likely to be formed relatively efficiently in the atmosphere
as they can originate from any unsaturated compound with a terminal double bond, but they
do not represent the entire Criegee intermediate population.
As SCI were found to react quickly with many trace gases, various model studies were
performed on the impact SCI have as oxidants in the atmosphere (Vereecken et al., 2012;Boy
et al., 2013;Percival et al., 2013;Pierce et al., 2013;Sarwar et al., 2013;Sarwar et al.,
2014;Novelli et al., 2014b;Vereecken et al., 2014). Some of these studies focused in
particular on the possible impact that SCI might have on the formation of sulfuric acid
($H_2SO_4$) in the gas phase, following Mauldin III et al. (2012) who suggested that Criegee
intermediates are the missing $SO_2$ oxidant needed to close the sulfuric acid budget over a
boreal forest. This is supported by theoretical and laboratory studies that have determined a
rate coefficient between SCI and sulfur dioxide ($SO_2$) of the order of $10^{-11}$ $cm^3$ $molecule^{-1}$ $s^{-1}$
(Aplincourt and Ruiz-López, 2000;Jiang et al., 2010;Kurtén et al., 2011;Vereecken et al.,
2012;Welz et al., 2012;Taatjes et al., 2013;Liu et al., 2014b;Sheps et al., 2014;Stone et al.,
2014). As the main atmospherically relevant oxidiser of $SO_2$ in the gas phase is the OH
radical ([OH] ~ 5 x $10^6$ molecules $cm^{-3}$) with a rather slow rate coefficient at ambient
temperature and pressure of 2 x $10^{-12}$ $cm^3$ $molecule^{-1}$ $s^{-1}$ (Atkinson et al., 2004), the high rate





coefficient for $SO_2$ oxidation would allow SCI to have a relevant impact on the $H_2SO_4$
formation even if present in small concentrations. The model studies have shown that,
depending on the environment, SCI can have a potentially important impact on $H_2SO_4$
formation. All these studies are affected by large uncertainties and many simplifications used
for coping with the paucity of data on the reactions of specific SCI with various trace gas
species, on the speciation of SCI, and on the steady state concentration of SCI in the
troposphere. Until now no direct or reproducible indirect method was able to determine the
steady state concentration of SCI in the lower troposphere.
In this paper, we firstly estimate the concentration of SCI in the lower troposphere, based on
the data collected during the HUMPPA-COPEC 2010 campaign (Williams et al., 2011) in a
Boreal forest in Finland and the HOPE 2012 campaign in rural southern Germany. The
budget of SCI is analyzed using four different approaches: 1) based on an unexplained $H_2SO_4$
production rate (Mauldin III et al., 2012); 2) from the measured concentrations of unsaturated
volatile organic compounds (VOC); 3) from the observed OH reactivity (Nölscher et al.,
2012); and 4) from an unexplained production rate of OH (Hens et al., 2014). Secondly, we
present measurements obtained using our inlet pre-injector laser-induced fluorescence assay
by gas expansion technique (IPI-LIF-FAGE) (Novelli et al., 2014a) during the HUMPPA-
COPEC 2010 and the HOPE 2012 campaigns. A recent laboratory study performed with the
same instrumental setup showed that the IPI-LIF-FAGE system is sensitive to the detection
of the OH formed from unimolecular decomposition of SCI (Novelli et al., 2014b). Building
on this study, the background OH ($OH_{bg}$) (Novelli et al., 2014a) measured during the two
field campaigns is investigated in comparison with many other trace gases in order to assess
if the observations in controlled conditions are transferable to the ambient conditions.



## 2 Instrumentation and field sites
### 2.1 IPI-LIF-FAGE description
A comprehensive description of the IPI-LIF-FAGE ground-based instrument, HORUS
(Hydroxyl Radical Measurement Unit based on fluorescence Spectroscopy), is given by
Novelli et al. (2014a) and only some important features of the instrument are highlighted
here. The IPI-LIF-FAGE instrument consists of: the inlet pre-injector (IPI), the inlet and
detection system, the laser system, the vacuum system and the instrument control and data
acquisition unit. The air is drawn through a critical orifice into a low pressure region (~300-
500 Pa) where OH molecules are selectively excited by pulsed UV light around 308 nm. The
light is generated at a pulse repetition frequency of 3 kHz by a Nd:YAG pumped, pulsed,
tunable dye laser system and is directed into a multipass "White cell" making 32 passes
through the detection volume (White, 1942). The air sample intersects the laser beam and the
fluorescence signal from the excited OH molecules is detected using a gated micro-channel
plate (MCP) detector. IPI, situated in front of the instrument inlet, is used to measure a
chemical zero to correct for possible internal OH signal generation. An OH scavenger is
added to the sample air 5 cm in front of the inlet pinhole in a concentration that allows a
known, high proportion of atmospheric OH to be scavenged (~ 90 %). The OH scavenger is
added every two minutes so that the instrument measures a total OH signal ($OH_{tot}$) when the
OH scavenger is not injected and a background OH signal ($OH_{bg}$) when the OH scavenger is
injected. The difference between these two signals yields the atmospheric OH concentration
($OH_{atm}$). The efficiency of this technique for measuring OH with this particular LIF-FAGE
instrument is described together with the IPI characterisation in Novelli et al. (2014a). The
OH calibration of the HORUS instrument is obtained via the production of a known amount





of OH and hydroperoxyl radicals ($HO_2$) from the photolysis of water at 185 nm using a
mercury lamp. A more detailed description of the instrument calibration is reported by
Martinez et al. (2010) and Hens et al. (2014). A calibration factor for the background OH
signal observed by the HORUS instrument is currently not available. Therefore, this signal
will be discussed and plotted in OH fluorescence counts per seconds (cps) measured by the
MCP, normalized by the laser power and corrected for quenching and sensitivity changes
towards the detection of OH. The sensitivity of the instrument towards the OH radical is
affected by: alignment of the white cell, optical transmission of the components, sensitivity of
the MCP, water vapor, internal pressure, and internal temperature (Martinez et al., 2010).
These factors affect the sensitivity of HORUS towards the background OH in a similar
manner as they mainly impact the sensitivity of the instrument to the detection of OH.
We hypothesise that the $OH_{bg}$ is formed chemically within the IPI-LIF-FAGE instrument.
Laser induced production of OH radicals was thoroughly tested in the laboratory and in the
field (Novelli et al., 2014a) showing that this background OH signal is not induced by the
laser beam from double pulsing, nor from air stagnating in the detection cell. By changing the
laser power, no quadratic dependency of the $OH_{bg}$ was observed even at night time, when the
contribution of the $OH_{bg}$ to the $OH_{tot}$ measured by the instrument is highest (Novelli et al.,
2014a). In addition, during the HUMPPA-COPEC 2010 and HOPE 2012 campaigns, the
correlation coefficient of the $OH_{bg}$ with the laser power was R = 0.002 and R = 0.2,
respectively.
In contrast, ozonolysis of alkenes performed during laboratory tests showed that the IPI-LIF-
FAGE instrument is sensitive to the OH formed from unimolecular decomposition of SCI
within the low pressure section of the instrument (Novelli et al., 2014b).





At present, titration of the $OH_{atm}$ was performed by most of the LIF-FAGE instruments
(Amédro, 2012;Mao et al., 2012;Griffith et al., 2013;Woodward-Massey et al., 2015;Griffith
et al., 2016;Tan et al., 2016) in different environments. Some of these instruments showed the
presence of an unknown interference (Mao et al., 2012;Griffith et al., 2013;Tan et al., 2016)
while for others no clear conclusions were drawn as a better operation of the titration unit is
needed (Amédro, 2012;Woodward-Massey et al., 2015). In addition, laboratory studies
(Fuchs et al., 2016;Griffith et al., 2016) have shown similarity with what was observed with
the IPI-LIF-FAGE during experiments of ozonolysis of alkenes although the origin of the OH
signal was not uniquely attributed to a particular mechanism.
Our hypothesis is that the $OH_{bg}$ measured in ambient air with the IPI-LIF-FAGE at least
partially originates from unimolecular decomposition of SCI. Section 4 describes the
observed behaviour of the signal during the campaigns and its relationship to other observed
chemical tracers and discusses if this is compatible with our hypothesis.

## 2.2   Measurement site and ancillary instrumentation

We present measurements from two sites, a boreal forest site in Finland and a rural site in
Southern Germany. The HUMPPA-COPEC 2010 (Hyytiälä United Measurements of
Photochemistry and Particles in Air – Comprehensive Organic Precursor Emission and
Concentration study) campaign took place during summer 2010 at the SMEAR II station in
Hyytiälä, Finland (61° 51' N, 24°17' E, 181 m a.s.l.) in a boreal forest dominated by Scot
Pines (*Pinus Silvestris L.*). The site hosts continuous measurements of several trace gases and
meteorological parameters as well as aerosol particles concentration, size distribution and
composition (Junninen et al., 2009). Further details and a more complete description of the





site, the instrumentation and the meteorological conditions during the campaign can be found
in Williams et al. (2011) and Hens et al. (2014). A brief description of the instruments used in
this study is given here. Ozone was measured by a UV photometric gas analyser (Model 49,
Thermo Electron Corporation). A gas chromatograph (GC, Agilent Technologies 6890A)
coupled to a mass-selective detector (MS, Agilent Technologies MSD 5973 *inert*) was used
for the measurements of BVOC (Yassaa et al., 2012). The total OH reactivity was measured
by the comparative reactivity method (CRM) (Sinha et al., 2008) for two different heights,
one within and one above the canopy (18 and 24 m, respectively) (Nölscher et al., 2012).
Sulfur dioxide ($SO_2$) concentration was measured with a fluorescence analyzer (Model 43S,
Thermo 20 Environmental Instruments Inc.). Aerosol number size distributions between 3 nm
and 950 nm were measured with a Differential Mobility Particle Sizer (DMPS) (Aalto et al.,
2001). The size distributions were used for calculating the loss rate of gas-phase sulfuric acid
via condensation sink (CS) with the method presented by Kulmala et al. (2001). Sulfuric acid
and OH radical concentrations were measured on the ground with the CIMS. The average
concentrations and their $1\sigma$ variability for the trace gases relevant for this study are listed in
Table 1. For the first period of the campaign, between the 27th and the 31st of July, the IPI-
LIF-FAGE instrument was run on the ground side-by-side with a chemical ionisation mass
spectrometer (CIMS; Petäjä et al. (2009)) measuring OH and sulfuric acid ($H_2SO_4$)
concentrations. On the 2nd of August the IPI-LIF-FAGE instrument was moved to the top of
the HUMPPA tower above the canopy and measured there for the remainder of the campaign
(12th of August). The data are therefore separated into ground and tower periods.
The HOPE 2012 (Hohenpeißenberg Photochemistry Experiment) campaign was conducted
during the summer of 2012 at the Meteorological Observatory in Hohenpeissenberg, Bavaria,
Germany (47° 48' N, 11° 2' E). The observatory is a Global Atmosphere Watch (GAW)





station operated by the German Meteorological Service (DWD) and is located at an altitude
of 985 m a.s.l. and about 300 m above the surrounding terrain, mainly consisting of meadows
and coniferous forests. More information about the site can be found in Handisides et al.
(2003). Ozone was measured by UV absorption with TEI 49C (Thermo Electron Corporation,
Environmental Instruments) (Gilge et al., 2010). Non-methane hydrocarbons (NMHC) were
measured with a GC-flame ionization detection (FID) system (series 3600CX, Varian,
Walnut Creek, CA, USA) (Plass-Dülmer et al., 2002). BVOC were detected using a GC
(Agilent 6890) with a FID running in parallel with a MS (Agilent Technologies MSD 5975
*inertXL*) described by Hoerger et al. (2014). Photolysis frequencies ($J(NO_2)$ and $J(O^1D)$) were
measured next to the IPI-LIF-FAGE with a set of filter radiometers (Handisides et al., 2003).
The OH reactivity was measured with two instruments for a short period of time from the 10[th]
until the 18[th] of July. One method was the CRM and the same instrument was used as during
the HUMPPA-COPEC 2010 campaign. The second method was a new application of the
DWD CIMS instrument (Berresheim et al., 2000) which also measured $H_2SO_4$ and OH
radicals.. As the data will be used only in a qualitative way for the current study, a very short
description of this novel technique is given here and details will be presented in a future
publication. With the CIMS instrument, OH radicals are measured by converting them into
$H_2SO_4$ after reaction with $SO_2$ in a chemical reactor and subtraction of a corresponding
background after scavenging the OH with propane (Berresheim et al, 2000). A second $SO_2$
titration zone was used 15 cm (or 140 ms) downstream of the first injection to determine the
OH decay from OH radicals generated in the UV-calibration zone immediately upstream of
the first titration. The difference between these two titration zones in two consecutive 2.5 min
intervals allows the determination of the OH decay, after correcting for ambient OH and wall
losses. The uncertainty is estimated at $\pm 2$ s$^{-1}$ and the limit of detection is 2 s$^{-1}$. $SO_2$





concentration was measured with a fluorescence analyzer and aerosol size distributions were
measured and used to calculate the loss rate of gas-phase sulfuric acid due to CS formed by
existing aerosol surface via the method presented by (Birmili et al., 2003). The average
concentrations and their 1σ variability for the trace gases relevant to this study are listed in
Table 1.
**3   SCI concentrations during HUMPPA-COPEC 2010 and HOPE 2012**
**3.1   Missing $H_2SO_4$ oxidant**
The study by Mauldin III et al. (2012) in a boreal forest during the HUMPPA-COPEC 2010
campaign showed a consistent discrepancy between the measured $H_2SO_4$ and the calculated
gas phase $H_2SO_4$ concentration when considering oxidation of $SO_2$ from OH radical and the
condensation onto pre-existing aerosol particles (CS, condensation sink) as the sole
production and loss processes, respectively (Eq. 1).
$$[H_2SO_4] = \frac{k_{OH+SO_2} \times [OH] \times [SO_2]}{CS} \qquad (1)$$
On average the sulfuric acid in the gas phase calculated using Eq. 1 was only half of the total
$H_2SO_4$ observed in the field and lied outside the uncertainties associated with the calculation
of the formation channel and the condensation sink (Mauldin III et al., 2012). Although no
unambiguous evidence links SCI to the missing oxidant, laboratory tests performed with a
similar instrument (Berndt et al., 2012;Berndt et al., 2014a;Sipilä et al., 2014) confirmed the
role that SCI could have in the oxidation of $SO_2$ and formation of $H_2SO_4$. Assuming that SCI
are the only other species in addition to OH that oxidize $SO_2$ in the gas phase, and knowing





1 the rate coefficient of SCI and OH with $SO_2$, it is possible to calculate the steady state

2 concentration of SCI in that environment:

3 $$[H_2SO_4] = \frac{(k_{OH+SO_2} \times [OH] + k_{SCI+SO_2} \times [SCI]) \times [SO_2]}{CS} \qquad (2)$$

4 The rate coefficient between OH and $SO_2$ at standard pressure is $(2.04 \pm 0.10)$ x

5 $10^{-12}$ $(T/300)^{-0.27}$ $cm^3$ molecule$^{-1}$ s$^{-1}$ (Atkinson et al., 2004). The rate coefficient of SCI with

6 $SO_2$ was determined by several groups, and the values cluster around two numbers. The first

7 one of ~ 5 x $10^{-13}$ $cm^3$ molecule$^{-1}$ s$^{-1}$ has been obtained by Mauldin III et al. (2012) and

8 Berndt et al. (2012) while another value of ~ $3.3 \pm 2$ x $10^{-11}$ $cm^3$ molecule$^{-1}$ s$^{-1}$ has been

9 obtained by a number of other groups (Welz et al., 2012;Taatjes et al., 2013;Liu et al.,

10 2014b;Sheps et al., 2014;Stone et al., 2014;Chhantyal-Pun et al., 2015;Newland et al.,

11 2015a;Newland et al., 2015b;Foreman et al., 2016;Zhu et al., 2016). Two explanations can be

12 put forward for this disagreement. The first is based on the fact that Mauldin III et al. (2012)

13 and (Berndt et al., 2012) measure the rate of formation of $H_2SO_4$ rather than directly the

14 reaction rate of SCI with $SO_2$. If, as suggested by Vereecken et al. (2012), the secondary

15 ozonide (SOZ) formed from the reaction between larger SCI and $SO_2$ can stabilize and

16 undergo bimolecular reaction without formation of $SO_3$, the difference in the rate coefficients

17 for the different experiments could be partly explained. However, more recent theoretical

18 work (Kuwata et al., 2015) found additional low-lying pathways that make collisional

19 stabilization of the SOZ unlikely. Experiments by Carlsson et al. (2012) and Ahrens et al.

20 (2014) observed high yields of $SO_3$ close to unity suggesting that the SOZ is not lost under

21 the conditions used, i.e. in chambers with high concentrations of reactants and in the absence

22 of water. At the same time, these reaction conditions differ from the other studies which were





performed either at ambient air conditions or with lower concentrations of reagents and in the
presence of water.
An alternative explanation could be based on analysis of the studies by Mauldin III et al.
(2012) and Berndt et al. (2012). In their experiments, the rate of the SCI+$SO_2$ reaction is
derived relative to the total loss rate of SCI, $L_{SCI}$, as it governs the steady-state concentration
of SCI with negligible $SO_2$ present. This $L_{SCI}$ has a value on the order of ~ 3 to 5 s$^{-1}$ in both
experiments. Since these studies, a large body of experimental and theoretical data has
become available, regarding the reactivity of SCI towards many coreactants present in the
reaction mixture (Taatjes et al., 2013;Ouyang et al., 2013;Ahrens et al., 2014;Buras et al.,
2014;Liu et al., 2014a;Stone et al., 2014;Sheps et al., 2014;Welz et al., 2014;Lewis et al.,
2015). From this new data, we should consider that a total loss rate of about 4 s$^{-1}$ is an
underestimate. In a previous study by Novelli et al. (2014b) a value of $L_{SCI}$ = 40 s$^{-1}$ under
atmospheric conditions was proposed. A re-analysis of the study by Mauldin III et al. (2012)
using $L_{SCI}$ = 40 s$^{-1}$ and the measured yield of SCI for α-pinene of 0.1 (Donahue et al., 2011),
results in a rate coefficient for the α-pinene-derived SCI + $SO_2$ reaction of $2.6 \times 10^{-11}$ cm$^3$
molecule$^{-1}$ s$^{-1}$. Likewise, for the other compounds examined in the two studies (Berndt et al.,
2012;Mauldin III et al., 2012), the derived rate of SCI+$SO_2$ would shift significantly towards
the higher values obtained in the other studies (Welz et al., 2012;Taatjes et al., 2013;Liu et
al., 2014b;Sheps et al., 2014;Stone et al., 2014). One must consider, though that the study by
Berndt et al. (2012) included a measurement of $k_{loss}$, based on the observed $H_2SO_4$ formation
from the steady state SCI in the absence of $SO_2$. Hence, this second explanation is only viable
if another source of $H_2SO_4$ exists in the system; this has already been suggested by Newland
et al. (2015a) based on their $SO_2$ oxidation experiments.





Still, as these explanations for the lower values by Mauldin III et al. (2012) and Berndt et al.
(2012) are merely speculative, we will consider both $3.3 \times 10^{-11}$ cm$^3$ molecule$^{-1}$ s$^{-1}$ and $5 \times$
$10^{-13}$ cm$^3$ molecule$^{-1}$ s$^{-1}$ as possible rate coefficients for the SCI + SO$_2$ reaction in the current
budget analysis.
The steady state concentration of SCI for the boreal forest was calculated using the measured
data and Eq. 2, yielding an average [SCI] = $(2.3 \pm 3) \times 10^4$ molecules cm$^{-3}$ for a $k_{SCI+SO_2}$ rate
coefficient of $3.3 \times 10^{-11}$ cm$^3$ molecule$^{-1}$ s$^{-1}$, and [SCI] = $(2 \pm 2) \times 10^6$ molecules cm$^{-3}$
obtained with $k_{SCI+SO_2}$ = $5 \times 10^{-13}$ cm$^3$ molecule$^{-1}$ s$^{-1}$. Note that both values for the steady
state concentration of SCI remain in agreement for polluted and pristine environments, 1.9 x
$10^6$ molecules cm$^{-3}$ and 4.5 x $10^4$ molecules cm$^{-3}$ respectively, as based on the concentrations
of measured VOC and O$_3$ (Welz et al., 2012).
A similar estimate of the SCI steady state concentration was derived for the HOPE 2012
campaign. The H$_2$SO$_4$ concentration during this campaign can be mainly explained by the
reaction between OH and SO$_2$. Figure 1 shows the correlation between the total production
rate of H$_2$SO$_4$ (P(H$_2$SO$_4$)$_{tot}$) calculated from the product of measured H$_2$SO$_4$ and the
condensation sink, as well as the production rate of H$_2$SO$_4$ from the reaction of OH and SO$_2$.
The linear regression following the method of York et al. (2004) yields a slope of 0.90 $\pm$ 0.02
with a negligible intercept (57 $\pm$ 7), accounting for the large uncertainty in the determination
of the CS (~ 30 %),. The steady state concentration of SCI was calculated from the measured
data and Eq. 2 yielding an average [SCI] of $(1 \pm 3)$ x $10^4$ molecules cm$^{-3}$ when using
$k_{SCI+SO_2}$ = 3.3 x $10^{-11}$ cm$^3$ molecule$^{-1}$ s$^{-1}$ and $(1 \pm 3)$ x $10^6$ molecules cm$^{-3}$ for $k_{SCI+SO_2}$ = 5 x
$10^{-13}$ cm$^3$ molecule$^{-1}$ s$^{-1}$.





### 3.2 Measured unsaturated VOC
Another method to estimate the SCI concentration is based on their production and loss
processes. In a forest SCI are expected to be formed from the ozonolysis of unsaturated
BVOC. It is possible to calculate an average steady state concentration for SCI using the
following equation

$$[SCI] = \frac{(\sum_i k_{VOC_i+O_3} \times [VOC_i]) \times [O_3] \times Y_{SCI}}{L_{SCI\,syn}}$$
(3)

Where $k_{VOC_i+O_3}$ is the rate coefficient between the $VOC_i$ and ozone (Table SI-1), $Y_{SCI}$ is the
yield of SCI in the ozonolysis reaction, and $L_{SCIsyn}$ is the total loss of *syn*-SCI. We obtain
[SCI] ≈ [SCI$_{syn}$] following the model described by Novelli et al. (2014b), which accounts for
many possible losses of SCI including the reaction with water dimers and unimolecular
decomposition. The latter study suggests that *anti*-acetaldehyde oxide and formaldehyde
oxide react quickly with water and water dimers and that their contributions can be neglected.
A yield of SCI formation ($Y_{SCI}$) of 0.4 was estimated based on the data by Hasson et al.
(2001). The steady state concentration of SCI for the HUMPPA-COPEC 2010 campaign was
calculated using the measured data for $[O_3]$ and $[VOC_i]$ and an average value of 40 s$^{-1}$
(Novelli et al., 2014b) for $L_{SCIsyn}$ as this value was found to be rather constant and mainly
dependent on the unimolecular decomposition rate of the SCI. Using Eq. 3 an average [SCI]
of ~ $(5 \pm 4) \times 10^3$ molecules cm$^{-3}$ is derived.
During the HOPE 2012 campaign a larger number of unsaturated organic trace gases, both
anthropogenic and biogenic, were measured (Table SI-1). For $Y_{SCI}$ the same value of 0.4 was
used while for $L_{SCIsyn}$ the value of 32 s$^{-1}$, obtained from the model described by Novelli et al.
(2014b) for the rural European environment, was used. Using these values in Eq. 3 results in





a steady state concentration of $[SCI] = (7 \pm 6) \times 10^3$ molecules cm$^{-3}$. It should be noted that
recent work on the unimolecular decomposition (Fang et al., 2016b;Long et al., 2016;Smith
et al., 2016) yields loss rate significantly faster than used here; this implies that the [SCI]
obtained here could be an overestimate.
**3.3  OH reactivity**
During HUMPPA-COPEC 2010, between 27[th] July and 12[th] August, an average OH
reactivity, R = 9.0 ± 7.6 s$^{-1}$, was measured. On average, the majority of the measured OH
reactivity ($R_{unex}$ = 7.4 ± 7.4 s$^{-1}$), 80 %, was not accounted for by the measured organic and
inorganic trace gases (Fig. SI-1).  Biogenic emissions comprised up to ~ 10 % of the total
measured OH reactivity and up to half of the calculated OH reactivity (Fig. SI-1). As the
measurement site was located in a pristine forest in environment, affected only little by
anthropogenic emissions (Williams et al., 2011), it is likely that a large fraction of the
unexplained OH reactivity was formed by unmeasured primary emissions by the vegetation
and secondary products of oxidation. By assuming that the unmeasured VOC are unsaturated,
and by using a lumped rate coefficient, $k_{VOC+OH,}$ between OH and the fraction of unspeciated
VOC of 7 x 10$^{-11}$ cm$^3$ molecule$^{-1}$ s$^{-1}$, typical for an OH addition to a carbon-carbon double
bond (Atkinson et al., 2004;Peeters et al., 2007), it is possible to estimate the concentration
[VOC$_{unknown}$] of VOC that would be necessary to close the OH reactivity budget (Eq. 4).
$$R_{unex} = k_{VOC+OH} \times [VOC_{unknown}]$$    (4)
Using Eq. 4 with the measured data, an average [VOC$_{unknown}$] of $(1 \pm 1) \times 10^{11}$ molecules cm$^{-3}$
is found. This value is substituted into Eq. 3 and a lumped rate for reaction of [VOC$_{unknown}$]
and O$_3$ of 7 x 10$^{-17}$ is used. This value is based on the rate coefficient of the measured VOC



with $O_3$ weighted with their abundance (Table SI-1). The same $Y_{SCI}$ and $L_{SCIsyn,}$ of 0.4 and 40
$s^{-1}$, respectively, were used as described in section 3.2. With these values a concentration of
SCI of ~ $(1 \pm 1)$ x $10^5$ molecules $cm^{-3}$ is obtained. This estimate contains larger uncertainties
compared to the previous estimates as the rate coefficient for ozonolysis of unsaturated
compounds varies by up to three orders of magnitude. In addition, the rate coefficient
between OH and unsaturated compounds, depending on whether these are unsaturated
NMHC or OVOC, primary emissions, or secondary oxidation products, varies by an order of
magnitude. A sensitivity study was done on the SCI estimates from the unexplained OH
reactivity to attempt to account for this uncertainty in rate coefficients. It is possible to
calculate a lower limit for the SCI concentration by using the highest rate coefficient between
OH and unsaturated compounds, 1 x $10^{-10}$ $cm^3$ $molecule^{-1}$ $s^{-1}$ (Atkinson et al., 2006) together
with a slow rate coefficient for the unsaturated compounds and ozone, 1 x $10^{-17}$ $cm^3$
$molecule^{-1}$ $s^{-1}$ (Atkinson et al., 2006), leading to a [SCI] = $(8.7 \pm 8.0)$ x $10^3$ molecules $cm^{-3}$.
For the upper limit, a slower rate coefficient for OH and unsaturated OVOC, ~ 3 x $10^{-11}$ $cm^3$
$molecule^{-1}$ $s^{-1}$ (Atkinson et al., 2006;Teruel et al., 2006) together with a higher rate coefficient
with $O_3$, 1 x $10^{-16}$ $cm^3$ $molecule^{-1}$ $s^{-1}$ (Atkinson et al., 2006) results in a concentration of [SCI]
= $(3 \pm 3)$ x $10^5$ molecules $cm^{-3}$. To these SCI concentration estimates, we add the SCI formed
from the measured unsaturated VOC, [SCI] = $(5 \pm 4)$ x $10^3$ molecules $cm^{-3}$, to obtain the total
SCI from all VOC.
During the HOPE 2012 campaign the total OH reactivity was on average $3.5 \pm 3.0$ $s^{-1}$. Using
the measured trace gas concentrations it is possible to calculate the expected OH reactivity
(Fig. SI-2). Table SI-2 lists all the species included in the calculation of the OH reactivity
with their rate coefficient with OH. An average value of $2.7 \pm 0.7$ $s^{-1}$ was calculated. Figure
SI-2 shows that half of the measured OH reactivity can be explained by inorganic compounds





which were present in higher concentrations compared to the HUMPPA-COPEC 2010
campaign (Table SI-2), methane and carbonyl compounds (mainly acetaldehyde and
propanal). On average, 24 % of the measured OH reactivity remains unexplained by the
measured trace gases. In contrast to the HUMPPA-COPEC 2010 campaign, in HOPE 2012 a
more complete speciation of VOC was measured (Table SI-1) and the site was influenced by
relatively fresh anthropogenic emissions. Despite this extensive speciation, the unexplained
OH reactivity could still be due to unmeasured VOC. By using the value of unexplained OH
reactivity of 0.8 s$^{-1}$ and proceeding as described above for the boreal forest environment, an
estimate $(1.0 \pm 0.2) \times 10^4$ molecules cm$^{-3}$ for the SCI concentration, with a lower and an
upper limit of $(2.9 \pm 0.7) \times 10^4$ molecules cm$^{-3}$ and $(1 \pm 0.2) \times 10^3$ molecules cm$^{-3}$,
respectively, can be estimated. To these SCI concentration estimates, we add the SCI formed
from the measured unsaturated VOC, $[SCI] = (7 \pm 6) \times 10^3$ molecules cm$^{-3}$, to obtain the total
SCI from all VOC.
**3.4   Unexplained OH production rate**
During the HUMPPA-COPEC 2010 campaign, the comprehensive measurements (Williams
et al., 2011) allowed the calculation of a detailed OH budget (Hens et al., 2014). Most of the
OH production during daytime is due to photolysis of $O_3$ and recycling of $HO_2$ back to OH
via reactions with NO and $O_3$. This result holds for both high (R > 15 s$^{-1}$) and low (R ≤ 15 s$^{-1}$)
OH reactivity episodes during the campaign. While the OH budget can be closed during
daytime (J(O$^1$D) > 3 $\times$ 10$^{-6}$ s$^{-1}$) for low OH reactivity periods, during periods with high OH
reactivity there was a large unexplained production rate of OH, $P_{OH}^{un\,explained} = (2 \pm 0.7) \times 10^7$
molecule cm$^{-3}$ s$^{-1}$, which can thus be surmised to originate from VOC chemistry. In addition,




for both periods, during night time ($J(O^1D) \leq 3 \times 10^{-6}$ s$^{-1}$), the IPI-LIF-FAGE and the CIMS
instruments both measured non-negligible OH concentrations (Hens et al., 2014) where most
of the OH production was from unknown sources ($P_{OH}^{un\,exp\,lained}$ = 1 ± 0.9 x 10$^6$ molecule cm$^{-3}$
s$^{-1}$ (1σ) and $P_{OH}^{un\,exp\,lained}$ = 1.7 ± 0.7 x 10$^7$ molecule cm$^{-3}$ s$^{-1}$ (1σ) for low and high reactivity,
respectively). Our hypothesis is that ozonolysis of VOC could represent the missing OH
source. Indeed, formation of OH from oxidation of unsaturated VOC has been shown to be an
important source of OH in winter, indoors and during night time (Paulson and Orlando,
1996;Geyer et al., 2003;Ren et al., 2003;Heard et al., 2004;Harrison et al., 2006;Johnson and
Marston, 2008;Shallcross et al., 2014). As OH formation from ozonolysis proceeds through
Criegee intermediates (Fig. 2), we can attempt to estimate a SCI concentration from the OH
budget.
Assuming that all unexplained OH production, $P_{OH}^{un\,exp\,lained}$ , comes from VOC ozonolysis with
a certain OH yield $Y_{OH}$ we obtain:
$$P_{OH}^{un\,exp\,lained} = k_{voc+O_3} \times [VOC_{unidentified}] \times [O_3] \times Y_{OH} \qquad (5)$$
where only the VOC not already included in the OH budget performed by (Hens et al., 2014)
are considered, i.e. the VOC causing the unknown OH reactivity discussed above. The
average total OH yield from ozonolysis, $Y_{OH}$, is estimated at about 0.6 based on observed OH
yields from the literature (Atkinson et al., 2006). OH formation from ozonolysis occurs
through two channels (Fig. 2): prompt formation by the decomposition of chemically
activated CI$^*$, and delayed OH by formation of SCI followed by their thermal decomposition;
there are also product channels not yielding OH. The prompt yield of OH, $Y_{OH}^{CI^*}$ is estimated





at ~ 0.4 from SCI scavenging experiments (Atkinson et al., 2004); the remaining yield $Y_{OH}^{SCI}$
is then formed from SCI, where $Y_{OH} = Y_{OH}^{CI^*} + Y_{OH}^{SCI}$ and hence $Y_{OH}^{SCI} \approx 0.2$ .
We adopt a value for $Y_{SCI}$ of 0.4, as argued in section 3.2. The SCI formed do not all
decompose to OH, e.g. *anti*-CI tend to form esters instead. We label all SCI able to yield OH
as $SCI_{syn}$, without mandating a speciation but following the observation that *syn*-CI usually
yield OH through the vinylhydroperoxide channel. The total SCI yield is then divided into a
fraction, $Y_{syn}$, forming $SCI_{syn}$, and the remainder, $Y_{anti}$, forming non-OH-generating SCI. Little
information is available on the $Y_{syn}$:$Y_{anti}$ ratio, with only a few theoretical calculations on
smaller alkenes and a few monoterpenes (Rathman et al., 1999;Fenske et al., 2000b;Kroll et
al., 2002;Nguyen et al., 2009b;Nguyen et al., 2009a). Across most of these compounds the
ratio of *syn*- to *anti*-SCI is always within a factor of 5, i.e. between 0.2 and 1.0 (Rickard et al.,
1999). Based on this, we estimate the ratio of $Y_{syn}$ to $Y_{anti}$ as 1:1. This number avoids
overestimating the impact of SCI in the OH production and, using the factor of 5 indicated
above, would cause a variation in the final [SCI] estimate of maximum 20 %, (see eq. 7 and
figure 3) well below the total uncertainty of the result.
The production of OH from $SCI_{syn}$ formed from VOC not included in the OH budget is then
$k_{OH} \times [SCI_{syn}]$ , where we estimate $k_{OH} \approx 20$ s$^{-1}$ as measured by Novelli et al. (2014b) for *syn*-
$CH_3CHOO$, and where the steady state concentration of the $SCI_{syn}$ , [$SCI_{syn}$], is determined by
the ratio of the formation processes and the sum $L_{SCIsyn}$ of the loss processes already defined
above:
$$\left[ SCI_{syn} \right] = \frac{k_{voc+O_3} \times [VOC_{unidentified}] \times [O_3] \times Y_{SCI} \times Y_{syn}}{L_{SCI_{syn}}}$$    (6)



Merging the above equations, expressing the measured OH production from unknown
sources as the sum of direct OH production from CI[*] and indirect from SCI$_{syn}$, we obtain:
$$P^{un\,explained} = k_{voc+O_3} \times \left[VOC_{unidentified}\right] \times [O_3] \times \left( Y_{OH}^{CI^*} + Y_{SCI} \times Y_{syn} \times \frac{k_{OH}}{L_{SCI_{syn}}} \right) \quad (7)$$
The measured $P_{OH}^{un\,explained}$ and [O$_3$], and the estimates of the other parameters allows us to
calculate the factor $k_{voc+O_3} \times$ [VOC$_{unidentified}$]. Substituting this factor into eq. 6 yields an
estimate of the steady state concentration of SCI$_{syn}$. With a value for $P_{OH}^{un\,explained}$ of 1 x 10$^6$
molecules cm$^{-3}$ s$^{-1}$ as observed for low reactivity episodes and at night during HUMPPA, a
steady state concentration of SCI$_{syn}$ of (2 ± 2) x 10$^4$ molecules cm$^{-3}$ is calculated. For high
reactivity episodes during HUMPPA-COPEC 2010, the missing $P_{OH}^{un\,explained}$ of 2 x 10$^7$
molecules cm$^{-3}$ s$^{-1}$ results in a SCI concentration of (4 ± 4) x 10$^5$ molecules cm$^{-3}$. To obtain
the total SCI concentration, we then need to add the non-OH-producing SCI. Here we assume
that these are mostly *anti*-SCI or H$_2$COO, both of which react rather quickly with H$_2$O or
(H$_2$O)$_2$ (Taatjes et al., 2013;Chao et al., 2015;Lewis et al., 2015), and that their contribution
can be neglected. We thus obtain that [SCI] ≈ [SCI$_{syn}$]. To this we add the SCI concentration
calculated from the measured unsaturated VOC (section 3.2), (5 ± 4) x 10$^3$ molecules cm$^{-3}$, to
obtain the SCI formed from all VOC.
For HOPE 2012 it is difficult to accurately derive an OH budget due to the lack of
information on the HONO concentration, which represents an important primary source of
OH. A detailed analysis of the OH production and loss during the campaign thus requires a
detailed model study to derive HONO concentrations, which is outside the scope of this




paper. Hence, an estimate on the SCI from a possible missing OH production rate during the
HOPE 2012 campaign is not included here.
Equation 7, for a given set of yields, unimolecular decomposition rates and SCI losses, allows
the estimate of the relative contribution of SCI and CI$^*$ to the total production rate of OH
from the ozonolysis of VOC. With the yields considered in this study and for a unimolecular
decomposition rate of SCI into OH of 20 s$^{-1}$, the SCI would contribute up to 12 % to the total
formation of OH from ozonolysis of VOC in both environments. This indicates that the SCI
do not have a large impact in the production of OH radicals and at the same time emphasizes
how important a realistic estimate of VOC concentration is for modeling the OH radical as
already underlined by (Hens et al., 2014).
**3.5    Robustness of the [SCI] estimates**
Figure 3 summarises the steady state concentration of SCI calculated on the basis of the
$H_2SO_4$ budget, the measured unsaturated VOC concentration and OH reactivity (R), and the
OH budget for the HUMPPA-COPEC 2010 and HOPE 2012 campaigns. By considering the
lower and the highest values estimated from the measured VOC and from the missing $H_2SO_4$
oxidant for both campaigns, respectively, the steady state concentration of SCI is calculated
to be between 5 x 10$^3$ and 2 x 10$^6$ molecules cm$^{-3}$ for the boreal forest environment during the
HUMPPA-COPEC 2010 campaign and between 7 x 10$^3$ and 1 x 10$^6$ molecules cm$^{-3}$ for rural
Germany during the HOPE 2012 campaign.  The SCI concentrations calculated using these
approaches represent a best-effort estimate made for the environments studied here based on
the available data; due to the many uncertainties related to the chemistry of SCI both in
production and loss processes these estimates span about two orders of magnitude.





The estimate of the SCI concentration from the sulfuric acid budgets relies on the rate of
oxidation of $SO_2$ to $H_2SO_4$. As indicated in section 3.1, two significantly different rate
coefficients for the reaction of SCI with $SO_2$ are currently available. One coefficient is high, ~
$3.3 \pm 2 \times 10^{-11}$ $cm^3$ molecule$^{-1}$ s$^{-1}$, while the other is several orders of magnitude lower, 5 x
$10^{-13}$ $cm^3$ molecule$^{-1}$ s$^{-1}$. Justifications of the differences in the values due to the diverse
procedures, i.e. direct detection of SCI + $SO_2$ for the high rate coefficient and detection of
$H_2SO_4$ for the lower one, are difficult, while recent measurements tend to agree with the
highest value. This casts doubts on the highest obtained SCI concentrations of ~ $10^6$
molecules cm$^{-3}$. In addition, the remaining three estimates strongly depend on the yield of
SCI, $k_{voc+o_i}$ and $L_{SCIsyn}$. Among these, the parameter with the highest uncertainty is the loss
rate of $syn$-SCI, $L_{SCIsyn,}$ as it is based on relatively few studies, which report large differences
between the observations. In this study, a value of 40 s$^{-1}$ and of 32 s$^{-1}$, based on previous
model analysis (Novelli et al., 2014b), for the HUMPPA-COPEC 2010 and HOPE 2012
campaigns respectively, were used. Recent work (Smith et al., 2016;Fang et al., 2016a;Long
et al., 2016) suggests a faster unimolecular decomposition rate for the acetone oxide Criegee
intermediate, exceeding $10^2$ s$^{-1}$ in ambient conditions. It is currently not clear whether this
rate applies to more substituted SCI as formed from monoterpenes but the use of these higher
decomposition rate in the model by (Novelli et al., 2014b) would result in a total $L_{SCIsyn}$ of ~
110 s$^{-1}$. This loss rate would decrease the estimated SCI concentration by almost a factor of 3,
closer to the lower estimates not exceeding $10^5$ molecule cm$^{-3}$; this also casts doubt on the
highest estimates given in figure 3. Therefore, a central estimate SCI concentration of about 5
x $10^4$ molecules cm$^{-3}$, with an order of magnitude uncertainty, is considered more appropriate
for both campaigns.





## 4    The source of the OH background signal

In this section we examine the background OH signal, $OH_{bg}$ (Novelli et al., 2014b) measured during the two field campaigns discussed in the previous sections. In particular, we examine if this signal is consistent with the SCI chemistry and concentrations indicated above.

### 4.1    Correlation of $OH_{bg}$ with temperature

The time series of the background OH signal measured during the HUMPPA-COPEC 2010 and HOPE 2012 campaigns are shown together with temperature and $J(O^1D)$ values in Fig. 4. Increases and decreases in the $OH_{bg}$ signal follow the temperature changes. During the HUMPPA-COPEC 2010 campaign the $OH_{bg}$ shows a strong correlation with temperature (Fig. 5) with a correlation coefficient $R = 0.8$ for the exponential fit. The exponential dependency with temperature is in agreement with data shown by Di Carlo et al. (2004) for the unexplained OH reactivity and indicates that the species responsible for the $OH_{bg}$ strongly correlate with emission of biogenic VOC (BVOC) such as monoterpenes and sesquiterpenes, which have been shown to also exponentially depend on temperature (Guenther et al., 1993;Duhl et al., 2008;Hakola et al., 2003). This suggests that $OH_{bg}$ is directly related to BVOC chemistry. The relationship between $OH_{bg}$ and temperature during the HOPE 2012 campaign is less obvious. It is possible to observe a weakly exponential correlation between the two ($R = 0.51$, Fig. SI-3) but there is very large scatter in the data. It is worthwhile to underline the differences between the two environments. The forest in Finland is essentially pristine and BVOC dominated while in southern Germany a larger fraction of non-biogenic VOC was observed. The lack of a clear exponential correlation between $OH_{bg}$ and temperature during the HOPE 2012 campaign could suggest different precursors or a different origin for the $OH_{bg}$ within the two environments.





During both campaigns a negligible correlation, R = 0.2, was observed between background
OH and $J(O^1D)$. This suggests that the $OH_{bg}$ does not primarily originates from photolabile
species.

## 4.2   Correlation of $OH_{bg}$ with unexplained OH reactivity

As described in section 3.3, during the HUMPPA-COPEC 2010 campaign high average OH
reactivity was observed (~ 9 $s^{-1}$), of which between 60 % and 90 %  cannot be explained by
the loss processes calculated from the measured species (Nölscher et al., 2012). A large
unexplained fraction of the reactivity has often been observed, especially in forested
environments (Di Carlo et al., 2004;Sinha et al., 2008;Edwards et al., 2013) indicating a large
fraction of undetected BVOC and/or secondary oxidation products. The $OH_{bg}$ shows some
correlation with the measured unexplained OH reactivity at 18 m, for the period on the
ground (R = 0.4), and the measured unexplained OH reactivity at 24 m, for the period on the
tower (R = 0.4) (Fig. 6). If  we consider only night time data, i.e. $J(O^1D) \leq 3 \times 10^{-6}$ $s^{-1}$ (Hens
et al., 2014), we obtain better agreement between the two dataset for both ground and tower
periods. During the night a large fraction of observed OH production (section 3.4) could not
be explained, which can tentatively be attributed to formation of OH from ozonolysis of
BVOC, suggesting that the background OH could be related to such a process. Correlation
between the $OH_{bg}$ and the OH reactivity was also observed in a study by Mao et al. (2012) in
a Ponderosa pine plantation (California, Sierra Nevada Mountains) dominated by isoprene
where even higher OH reactivity was observed (~ 20 $s^{-1}$).
During the HOPE 2012 campaign such a correlation with the unexplained OH reactivity was
not observed (R = 0.1). The OH reactivity was, on average, 3 times less than during the
campaign in Finland and, as shown in section 3.3, of which 50 % can be explained by
reaction of OH with methane, formaldehyde, acetaldehyde, inorganic compounds (NOx, $SO_2$,





CO) and anthropogenic VOC. On average only 17 % of the OH reactivity is caused by
reaction of OH with BVOC in this environment (Fig SI-2), dropping to 10 % during the
night. The unexplained OH reactivity is not influenced by distinguishing between day and
night time data suggesting a small contribution of non-measured BVOC. As this site is more
strongly affected by anthropogenic emissions (Table SI-2) compared to the site in Finland,
assuming that the $OH_{bg}$ originates from BVOC driven chemistry, a lack of correlation
between $OH_{bg}$ and OH reactivity can be expected.

### 4.3    Correlation of $OH_{bg}$ with ozonolysis chemistry

During the HUMMPA-COPEC 2010 campaign a high correlation with $O_3$, R = 0.7 (Fig. SI-
4), indicates that background OH likely originates from ozonolysis processes. A comparison
of background OH with the product of ozone concentration, measured unsaturated VOC
concentration and their ozonolysis rate coefficient does not show the same relationship. No
correlation (R = 0.05) is found by using the measured BVOC concentrations (Table SI-1). As
most of the OH reactivity remains unexplained, with measured BVOC comprising less than
10 % of the measured OH reactivity (Fig SI-1, Table SI-2), the lack of correlation could
suggest that the VOC responsible for the formation of SCI detected by the HORUS
instrument are likely part of the large fraction of unmeasured species to which a correlation
was reported in the previous section.
During HOPE 2012 a weak correlation was observed between background OH and ozone (R
= 0.5, fig. SI-5). This campaign, from July $10^{th}$ to August $19^{th}$ 2012, encompasses a time
period, from $1^{st}$ to $3^{rd}$ of August 2012, which was characterized by tree cutting in the vicinity
of the measurement site. During this period a significantly larger fraction of unexplained OH





reactivity, up to 40 % (Fig. SI-6), was observed. This suggests the presence of unidentified
BVOC emitted from the trees as a result of the stress induced on the plants from the cutting
activity, yet the concentrations of the BVOC listed in table SI-1 did not show any particular
increase during the same period. Figure 7 shows the correlation between $OH_{bg}$ and the
product $k_{O3}[VOC][O_3]$ of measured unsaturated VOC concentration (Table SI-1), $[O_3]$ and
the relevant ozonolysis rate coefficients. In red are depicted the data points belonging to the
tree cutting period, which naturally correspond to a larger $OH_{bg}$ concentration for similar
concentrations of measured VOC during the rest of the campaign, as the additional
contribution from the non-identified BVOC is neglected. The overall correlation appears to
be pretty poor in particular due to the few points scattering in the lower right corner. These
points all belong to three consecutive days, from 26[th] to 28[th] of July, which were
characterised by high temperature and large concentrations of BVOC (Table SI-3). As
noticeable in figure 4, during those three days the $OH_{bg}$ strongly deviates from the
temperature trends and reaches lower values. At present, the reason for such a low
concentration of $OH_{bg}$, during a period which should favour its formation if it originates from
SCI, is unclear. The instrument was left unattended at the site and the drop in the quality of
the signals required its shutdown on the evening of the 28[th] of July. However, as no evidence
was found to suggest an error in the data the points have not been discarded. Excluding that
period yields a correlation factor of R = 0.65. The correlation line intercept could arise for a
number of reasons. Unmeasured components of the OH reactivity (i.e. unspeciated VOCs)
are not accounted for in the calculation, and accounting for this unmeasured reactivity in this
calculation would shift the data to higher [VOC], decreasing the positive intercept. This is
also consistent with a higher intercept for the cutting tree period where a larger unexplained
OH reactivity was observed. It is also conceivable that the intercept is in part due to an





additional, non-ozonolysis source of background OH. One candidate for the night time
periods could be $NO_3$ as found in the work by Fuchs et al. (2016). Unfortunately, there was no
measurement of the $NO_3$ radical during the HOPE 2012 campaign, but based on previous
studies (Handisides et al., 2003), a concentration up to 14 pptv of $NO_3$ could be present and
could have a detectable impact.
Apart from the possible partial origin of $OH_{bg}$ from a $NO_3$ or other interferences, there are
also indications that the background OH could originate from ozonolysis of unsaturated
biogenic compounds. The correlation analysis requires that all VOCs are accounted for, and
omitting large contributions from unspeciated VOCs, as evidenced e.g. by OH reactivity
measurements, can be expected to reduce the correlation as observed in the case of
HUMPPA-COPEC 2010. The lack of correlation during the period from 26[th] to 28[th] July
2012 during HOPE-2012 characterised by large BVOC emissions remains unclear.

## 4.4   Correlation of $OH_{bg}$ with $P(H_2SO_4)_{unex}$

During both campaigns, measurements of $H_2SO_4$, $SO_2$, OH and CS (condensation sink) were
performed allowing the calculation of the sulfuric acid budget in the gas phase. As shown by
Mauldin III et al. (2012), during the HUMPPA-COPEC 2010 campaign the well-known $SO_2$
oxidation process by OH (Wayne, 2000) (Eq. 1) was not sufficient to explain the measured
concentration of $H_2SO_4$. As shown in section 3.1, half of the production rate of $H_2SO_4$, ~ 1 x
$10^4$ molecules $cm^{-3}$ $s^{-1}$, cannot be explained by reaction with OH radicals (Fig. 8). The
missing oxidant is assumed to be SCI, as discussed in section 3.1, because of their fast
reaction rate with $SO_2$. As our hypothesis about the origin of the $OH_{bg}$ supports this
assumption, we compared the $[H_2SO_4]_{unex}$ observed during the HUMPPA-COPEC 2010





campaign with the $OH_{bg}$ multiplied by $SO_2$ for the ground-based period when the instruments
(HORUS and CIMS) measured side-by-side (Fig. 9). The two datasets indicate a correlation
coefficient of R = 0.6 suggesting that, whichever species is responsible for the oxidation of
$SO_2$, is related to the formation of OH within the HORUS instrument.
Note that for the HOPE 2012 campaign the same budget calculation shows only a small
fraction (10 %) of unexplained $H_2SO_4$ production rate (Fig. 2).
Assuming SCI to be the unknown $SO_2$ oxidant, the results observed in both campaigns are in
agreement with the modeling study by Boy et al. (2013), who analyzed measurements at the
same sites described in this study. Similar to our result, they found a larger contribution of
SCI in the formation of $H_2SO_4$ for the boreal forest compared to rural Germany. As the OH
concentration differs by, on average, less than 50 % between the two environments, a similar
concentration of SCI in HOPE to that calculated for HUMPPA-COPEC 2010 would
contribute up to 30 % in the formation of $H_2SO_4$. However, the $H_2SO_4$ budget during this
campaign can approximately be closed by only considering the measured OH concentrations,
suggesting that the concentration of SCI in this environment is smaller than that during the
HUMPPA-COPEC 2010 campaign. This is consistent with the calculation in section 3 based
on the smaller reactivity and hence smaller VOC concentration in this environment
**4.5   Scavenging experiments**
A series of scavenging tests of the $OH_{bg}$ was performed during the HOPE 2012 campaign to
help identify the interfering species. $SO_2$ was chosen as scavenger for the species causing the
$OH_{bg}$, as it has been shown in several laboratory studies to react quickly with SCI (k ~ 3.3 x
$10^{-11}$ $cm^3$ $molecule^{-1}$ $s^{-1}$) mostly independently of their structure (Taatjes et al., 2014). The
injection of $SO_2$ was performed through the IPI system (Novelli et al., 2014a) together with
an OH scavenger. First the OH scavenger propane was injected within IPI to remove the





atmospheric OH; subsequently, $SO_2$ was injected in addition to the OH scavenger (Fig. 10).
The concentration of $SO_2$ is small enough not to scavenge SCI inside the low pressure section
of the instrument, nor is it additionally removing atmospheric OH within the IPI system as
the lifetime of OH by reaction with $SO_2$ is 200 times than of propane. With the addition of
$SO_2$ (1 x $10^{13}$ molecules cm$^{-3}$ in the sampled air) it is possible to suppress the $OH_{bg}$ signal
from the instrument to within the zero noise, indicative that the $OH_{bg}$ signal originates from
an SCI-like species that reacts with $SO_2$ and decomposes unimolecularly to OH.

## 4.6   SCI as a source of background OH

During the HUMPPA-COPEC 2010 campaign the background OH showed a strong
exponential relationship with temperature (R = 0.8) and it correlates with unexplained OH
reactivity (R = 0.5), which suggests correlation with BVOC, with ozone (R = 0.7), and also
with the $P(H_2SO_4)_{unex}$ (R = 0.6). During the HOPE 2012 campaign a weak exponential
correlation with temperature was recognized (R = 0.5) but no correlation was observed with
OH reactivity. The $OH_{bg}$ correlated with the product of ozone and unsaturated VOC for most
of the campaign (R = 0.6) although not for a period of three days at the end of July with
partly higher BVOC-$O_3$ turnover. In addition, during HOPE 2012 the $OH_{bg}$ signal was
scavenged by the addition of $SO_2$.
All evidence presented indicates that substantial parts of the $OH_{bg}$ originates from a species
formed during the ozonolysis of unsaturated VOC that decomposes into OH, is removable by
$SO_2$ and, if present in a significant concentration, increases the $H_2SO_4$ production. We are
currently not aware of any chemical species, other than SCI, known to oxidise $SO_2$ at a fast
enough rate and also decompose into OH. In addition, HORUS was shown to be sensitive to



the OH formed after unimolecular decomposition of SCI in the low-pressure region of the
instrument (Novelli et al., 2014b) in controlled laboratory studies. During the HUMPPA-
COPEC 2010 campaign, the correlation with OH reactivity improved when considering only
data during night time, the period during which a higher fraction of the production rate of OH
could not be accounted for (Hens et al., 2014). Indeed, during the night recycling via
$HO_2 + NO$ is low due to the negligible NO concentration, therefore a different path of
formation of OH is expected. One likely path could be the formation of OH from excited and
stabilised CI formed from ozonolysis of unsaturated compounds.
The considerations above are all consistent with the hypothesis that $OH_{bg}$ largely originates
from unimolecular decomposition of SCI in the field as well as in the laboratory.
Attempts to analyse the absolute concentration of SCI based on our $OH_{bg}$, however, indicates
that this hypothesis is not without difficulties. A particular problem is that to date no method
is available to produce and quantify a known concentration of a specific SCI conformer,
which precludes the absolute calibration of SCI-generated OH. *A priori*, it seems unlikely
that the IPI-LIF-FAGE instrument calibration factor for ambient OH, i.e. sampled from
outside the instrument through the nozzle, is identical to the sensitivity for OH generated
inside. The transmission factor through our nozzle pinhole is currently not known for OH
radicals; the calibration factor used for ambient OH accounts for this transmission as well as
for e.g. OH losses on the walls, alignment of the white cell, transmission optics, and response
of the MCP. These last three factors should affect the OH generated from any interfering
species similarly, while wall losses and transmission through the pinhole are different and
possibly also differ between SCI conformers. Additionally, different SCI vary in their
unimolecular decomposition rates and hence affect calibration by a different time-specific
OH yield. For example, theoretical studies (Vereecken and Francisco, 2012) and laboratory





experiments (Smith et al., 2016) indicate that acetone oxide will decompose faster than *syn*-
acetaldehyde oxide causing the formation of a different amount of OH, which in turn will
also be affected by different loss rates in the low pressure segment of the instrument. Thus, it
is not possible to convert the internal OH to an absolute SCI concentration since the mixture
of SCI is not known. At best one could obtain an "average" sensitivity factor, if one knew the
$OH_{bg}$ formed from a series of reference SCI conformers, and if the ambient SCI speciation is
known and not too strongly dependent on reaction conditions. To further illustrate the need of
a SCI-specific calibration, we try to simply calculate the external [SCI] from the internal
$OH_{bg}$ signal strength, calibrated based on the combined experimental and modelling study by
Novelli et al. (2014b). For a SCI mixture that behaves identical to syn-$CH_3CHOO$, the $OH_{bg}$
from the HUMPPA-COPEC 2010 campaign would then indicate an external [SCI] $\geq 2 \times 10^7$
molecules $cm^{-3}$, well above the estimates presented in section 3. Moreover, the observed
$OH_{bg}$ signal interpreted in this way would imply an ambient OH production exceeding $4 \times$
$10^8$ molecules $cm^{-3} s^{-1}$, clearly in disagreement with known chemistry, and also inconsistent
with our estimates (Fig 1). If we assume a faster decomposition rate for the SCI of 200 $s^{-1}$, a
higher fraction of the SCI decomposes in the low-pressure region, i.e. 80 % compared to 25
% for $k_{uni} = 20 s^{-1}$. This leads to a higher OH signal per SCI, and from this a [SCI] of $4 \times 10^6$
molecules $cm^{-3}$, though the implied ambient OH production would remain significantly too
high. Thus, the conversion of the OH signal to an absolute concentration of ambient SCI is
not unambiguous without full SCI speciation and knowledge of their chemical kinetics. Note
furthermore that these [SCI] estimates would represent a lower limit as we only observe SCI
that decompose to OH, whereas e.g. *anti*-SCI convert to acids/esters.
In an effort to work towards SCI-specific calibration, we probed the transmission of OH and
*syn*-$CH_3CHOO$ through the nozzles and the low-pressure region in the instrument, with





explorative laboratory tests using a traditional nozzle and a molecular beam skimmer nozzle.
The difference between these two nozzles is shown in Figure 11: the traditional nozzle, with a
pinhole drilled through the tip of a cone, is characterized by quick gas expansion in the area
immediately below the pinhole, contacting the wall surface of the bore. During ambient
measurements, deposition of molecules in position A in Figure 11 has been observed, further
illustrating the prevalence of wall contact. The molecular beam skimmer nozzle, on the other
hand, has much thinner sidewalls and a significantly narrower gas expansion, strongly
reducing wall contact. The laboratory test showed that the OH radical has a 23 % higher
transmission through the molecular beam nozzle compared to the traditional nozzle. The *syn*-
acetaldehyde oxide did not show any statistical difference in the transmission between the
two nozzles. This indicates that (a) SCI and OH have a different transmission efficiency and
most likely different wall losses, underlining that the OH calibration factor is not applicable
to SCI for ambient measurements, and (b) that the calibration factor for OH obtained for
ambient OH alone does not allow the quantification of the absolute OH concentration in the
low-pressure section of the FAGE instrument. This is the fundamental reason why the earlier
simple estimate of [SCI] and OH production leads to strong over-estimations.
In addition to the above effects, one should also consider that OH-production from SCI in the
low-pressure section might be catalysed to proceed at rates beyond their ambient counterpart,
biasing our interpretation of their ambient fate. The catalysis might involve wall-induced
isomerisation of the higher-energy *anti*-SCI to the more stable, OH-producing *syn*-SCI,
which would artificially increase the *syn*:*anti* ratio. Another possibility is the evaporation of
clusters stabilizing the SCI, as it is known that SCI efficiently form complexes with many
compounds, including water, acids, alcohols, hydroperoxides, $HO_x$ radicals, etc. (Vereecken
and Francisco, 2012). Redissociation of secondary ozonides (SOZ) seems less important,




except perhaps the SOZ formed with $CO_2$ (Aplincourt and Ruiz-López, 2000), which has no
alternative accessible unimolecular channels. At present, insufficient (if any) information is
available to assess the impact of such catalysis.
Taking into account the factors considered above, and assuming that the estimates for the SCI
concentration in both environments are correct, it appears unlikely that SCI are responsible
for such a large $OH_{bg}$ signal as observed by the HORUS instrument. If SCI were to be solely
responsible for the $OH_{bg}$ signal, the HORUS instrument would need to be far more sensitive
to the detection of SCI than to the detection of OH radicals by, for example, pinhole losses
that are 100 times smaller for SCI than for OH radicals. The evident discrepancy between the
qualitative evidence in support of the SCI hypothesis and the current quantitative difficulty in
reconciling the $OH_{bg}$ signal with the estimated ambient concentration of SCI does not allow
an unequivocal identification of the origin of the $OH_{bg}$ within our system. It cannot be
excluded that multiple species are contributing to the $OH_{bg}$ signal. $NO_3$ chemistry during
night time has been identified as a possible source of $OH_{bg}$ in the LIF-FAGE instrument of
the FZ-Jülich (Fuchs et al., 2016). However, in the case of the large observed night time $OH_{bg}$
concentrations during HUMPPA-COPEC 2010, the measured night time $NO_3$ concentrations
was below 1 ppt and therefore too small to explain the observed $OH_{bg}$.
**5   Conclusions**
We estimated a steady state concentration of SCI for the HUMPPA-COPEC 2010 and the
HOPE 2012 campaigns based on a large dataset. Starting from four different approaches, i.e.
based on unaccounted (i.e. non-OH) $H_2SO_4$ oxidant, measured VOC concentrations,
unexplained OH reactivity or unexplained production rates of OH, we estimated the





concentration of SCI to be between $\sim 10^3$ and $\sim 10^6$ molecules $cm^{-3}$. The highest values in
this range are linked to an assumed low rate coefficient for SCI + $SO_2$ of 5 x $10^{-13}$ $cm^3$
molecule$^{-1}$ s$^{-1}$ (see section 3.1), which is at odds with a larger body of more direct
measurements on this rate coefficient. Hence, higher values appear to be relatively less likely.
We thus obtain a central estimate SCI concentration of about 5 x $10^4$ molecules $cm^{-3}$, with an
order of magnitude uncertainty, for both campaigns. At such concentrations, SCI are
expected to have a significant impact on $H_2SO_4$ chemistry during the HUMPPA-COPEC
2010 campaign while during the HOPE 2012 campaign their impact is much smaller and
possibly negligible. Additionally, it was shown that, based on the yields and unimolecular
decomposition rate applied in this study, SCI do not have a large impact on the OH
production compared to the direct OH generation from ozonolysis of unsaturated VOC.
During both campaigns, the IPI-LIF-FAGE instrument detected an OH background signal
that originates from decomposition of one or more species inside the low pressure region of
the instrument. The source compound of the $OH_{bg}$ was shown to be unreactive towards
propane but to be removed by $SO_2$, and a relationship was found with the unaccounted $H_2SO_4$
production rate. It correlates with temperature in the same way as the emission of terpenes
and, in most but not all measurements periods, with the product of unsaturated VOC and
ozone as well as with the OH reactivity. While it is not possible at the moment to
unequivocally state that $OH_{bg}$ originates from stabilised Criegee intermediates, the
observations are consistent with known SCI chemistry. The contribution of SCI to the
observed $OH_{bg}$ cannot be quantified until a calibration scheme for SCI in the IPI-FAGE
system has been developed.
The predicted SCI concentrations derived in this study are low, likely not exceeding $10^5$
molecule $cm^{-3}$, therefore, the presence of SCI is unlikely to have a large impact on





atmospheric chemistry; the main exception appears to be $H_2SO_4$ production in selected
environments.

## Acknowledgements

LV was supported by the Max Planck Graduate Centre with the Johannes Gutenberg-
Universität Mainz (MPGC).
Work during HUMPPA-COPEC was supported by the Hyytiälä site engineers and staff.
Support of the European Community Research Infrastructure Action under the FP6
"Structuring the European Research Area" Programme, EUSAAR Contract No RII3-CT-
2006-026140 is gratefully acknowledged. The HUMPPA-COPEC 2010 campaign
measurements and analyses were supported by the ERC Grant ATMNUCLE (project No
227463), Academy of Finland Centre of Excellence program (project No 1118615), ,
Academy of Finland Centre of Excellence in Atmospheric Science – From Molecular and
Biological processes to The Global Climate' (ATM), 272041, the European integrated project
on Aerosol Cloud Climate and Air Quality Interactions EUCAARI (project No 036833-2),
the EUSAAR TNA (project No 400586), and the IMECC TA (project No 4006261).
The work during HOPE 2012 was supported by the scientists and staff of DWD
Hohenpeißenberg whom we would like to thank for providing the "platform" and opportunity
to perform such campaign. In particular, we thank, Anja Werner, Jennifer Englert and Katja
Michl for the VOC measurements, Stephan Gilge for the trace gases measurements and
Georg Stange for running the CIMS.
We also would like to thank Markus Rudolf for much technical support and guidance, Eric
Regelin and Umar Javed for the numerous scientific discussions Petri Keronen for providing
meteorological and trace gas concentration data from Hyytiälä during the HUMPPA-COPEC
2010 and Thorsten Berndt for providing the data to re-evaluate the rate coefficient between
SCI and $SO_2$.



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



Table 1. Average concentration (molecule cm$^{-3}$), with 1σ variability, of trace gases relevant for this
study.

| Compound | HUMPPA-COPEC 2010 | HOPE 2012 |
|---|---|---|
| SO$_2$[a] | $(1.4 \pm 1.7)$ x $10^{10}$ | $(2.2 \pm 2.3)$ x $10^9$ |
| H$_2$SO$_4$[a] | $(2 \pm 2)$ x $10^6$ | $(8.5 \pm 8.5)$ x $10^5$ |
| OH[a] | $(7 \pm 8)$ x $10^5$ | $(1.6 \pm 1.6)$ x $10^6$ |
| O$_3$[a] | $(1.1 \pm 0.2)$ x $10^{12}$ | $(1.1 \pm 0.3)$ x $10^{12}$ |
| Σ[VOC][a,b] | $(7.3 \pm 7.1)$ x $10^9$ | $(9.8 \pm 9.0)$ x $10^9$ |
| OH Reactivity | $9.0 \pm 7.6$ [c] | $3.5 \pm 3.0$ [c] |
| Condensation sink (CS) | $(10 \pm 4.0)$ x $10^{-3}$ [c] | $(7 \pm 3)$ x $10^{-3}$ [c] |

b,  HUMPPA COPEC 2010: isoprene, (−)/(+) α-pinene, (−)/(+) β-pinene, 3-carene, and
4        myrcene.
5        HOPE 2012: isoprene, α-pinene, β-pinene, 3-carene, myrcene, limonene, 2-
6        methylpropene, but-1-ene, sabinene, γ-terpinene, propene, cis-2-butene and ethene.
c,  Units: s$^{-1}$.



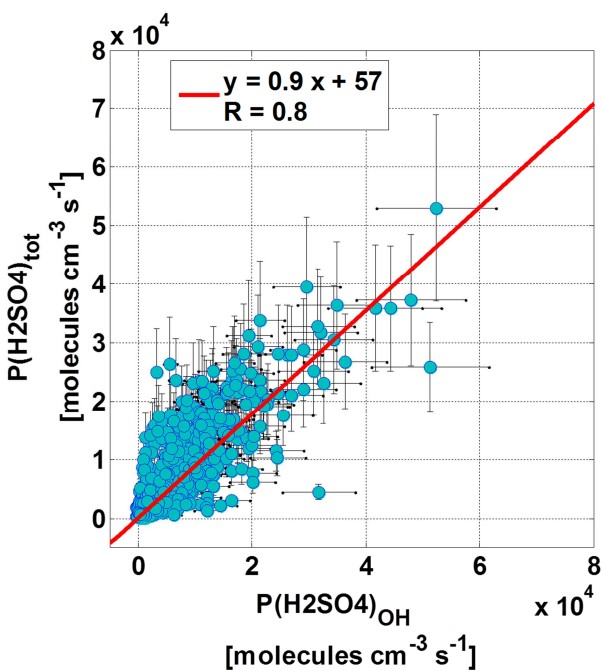

Figure 1. Total production rate of $H_2SO_4$ ($P(H_2SO_4)_{tot}$) as a function of the production rate of
$H_2SO_4$ from the reaction between OH and $SO_2$ during the HOPE 2012 campaign. The linear
regression, following the method of York et al. (2004), yields a slope of $0.9 \pm 0.02$ and a
intercept of $57 \pm 7$.





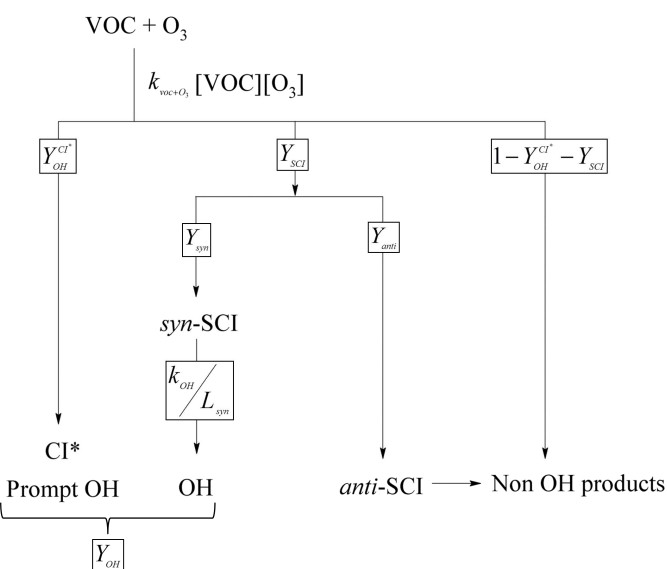

Figure 2. Schematic representation of the formation of OH from the ozonolysis of unsaturated
VOC.




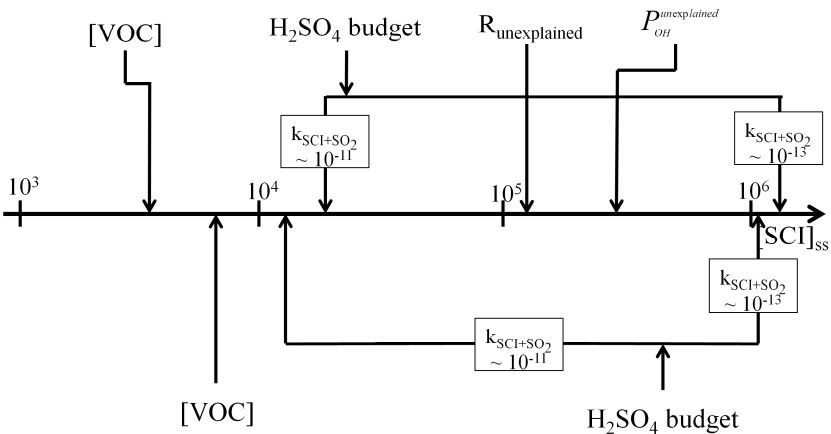

Figure 3. Schematic overview of the estimated steady state concentration of SCI ($[SCI]_{ss}$, molecules $cm^{-3}$) observed during the HUMPPA-COPEC 2010 and HOPE 2012 campaigns. For both campaigns the SCI estimate is based on the unsaturated VOC concentration measured, [VOC], and the $H_2SO_4$ budget using different $SCI+SO_2$ rate coefficients ($k_{SCI+SO2}$ in $cm^3$ $molecule^{-1}$ $s^{-1}$). In addition, during the HUMPPA-COPEC campaign SCI can be calculated from the unexplained OH reactivity, $R_{unexplained}$, and unexplained OH production, $P_{unexplained}^{OH}$. See main text for more details (Section 3).



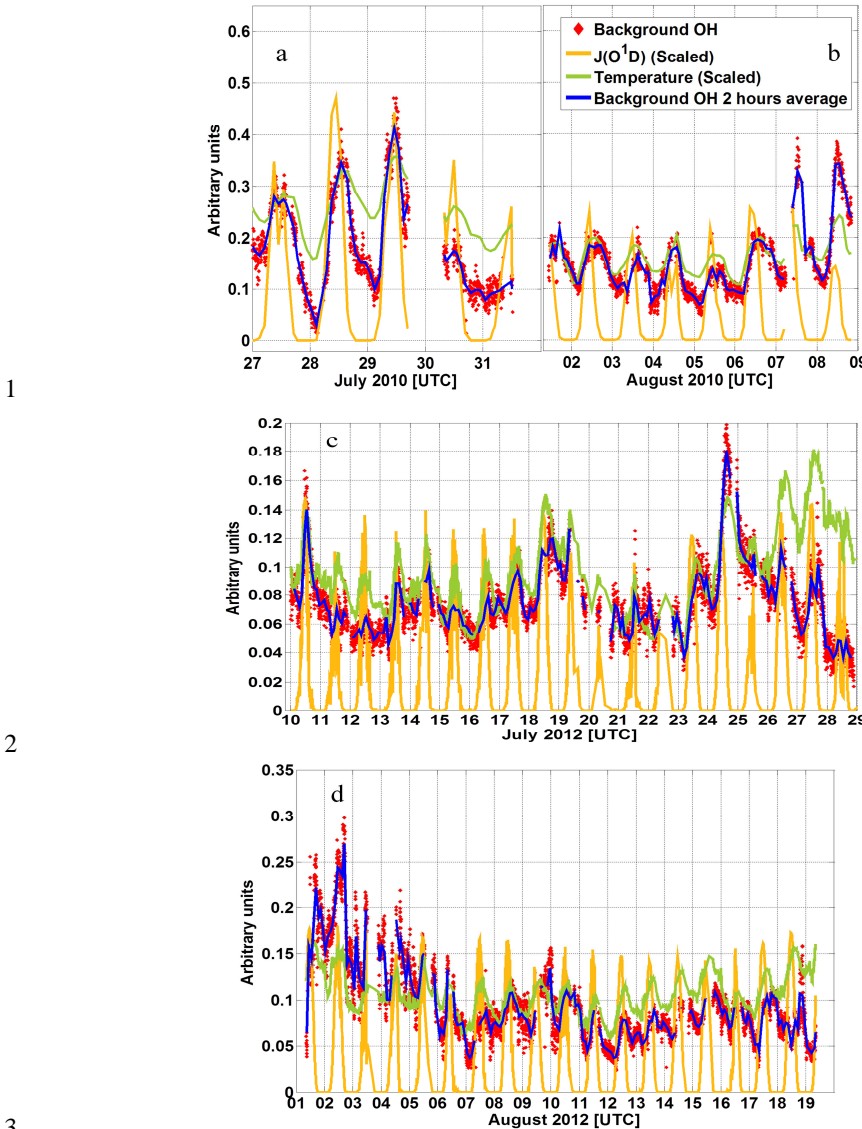

Figure 4. Background OH (red diamonds) measured during the HUMPPA-COPEC 2010 (a,
ground and b, tower) and the HOPE 2012 (c, July and d, August) campaigns together with
scaled $J(O^1D)$, multiplied by $4 \times 10^4$ and $4 \times 10^3$ for HUMPPA-COPEC 2010 and HOPE
2012, respectively (orange), and scaled temperature divided by 90 and 160 K for HUMPPA-
COPEC 2010 and HOPE 2012, respectively (green).





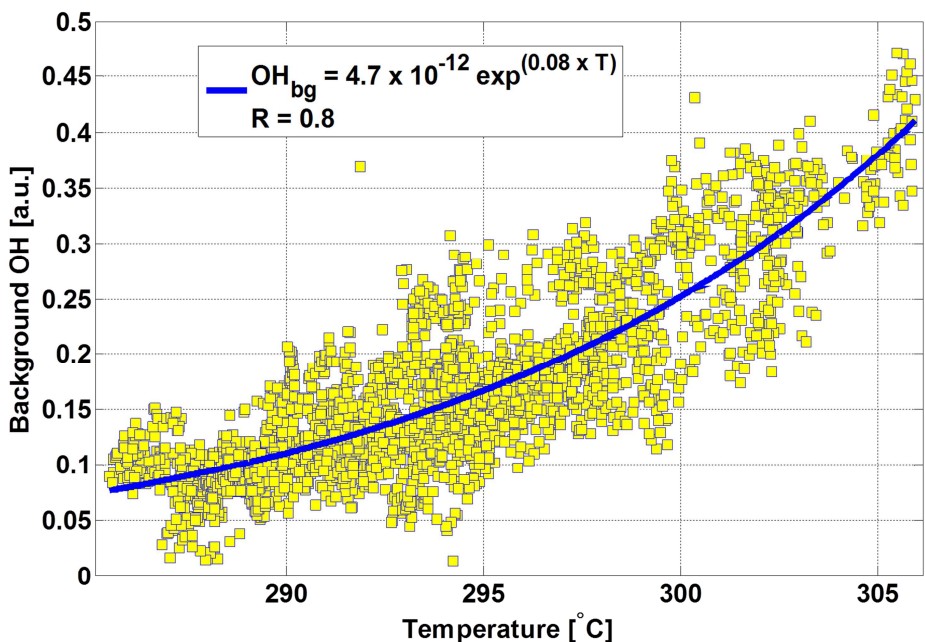

2 Figure 5. Background OH as a function of the temperature during the HUMPPA-COPEC
3 2010 campaign.



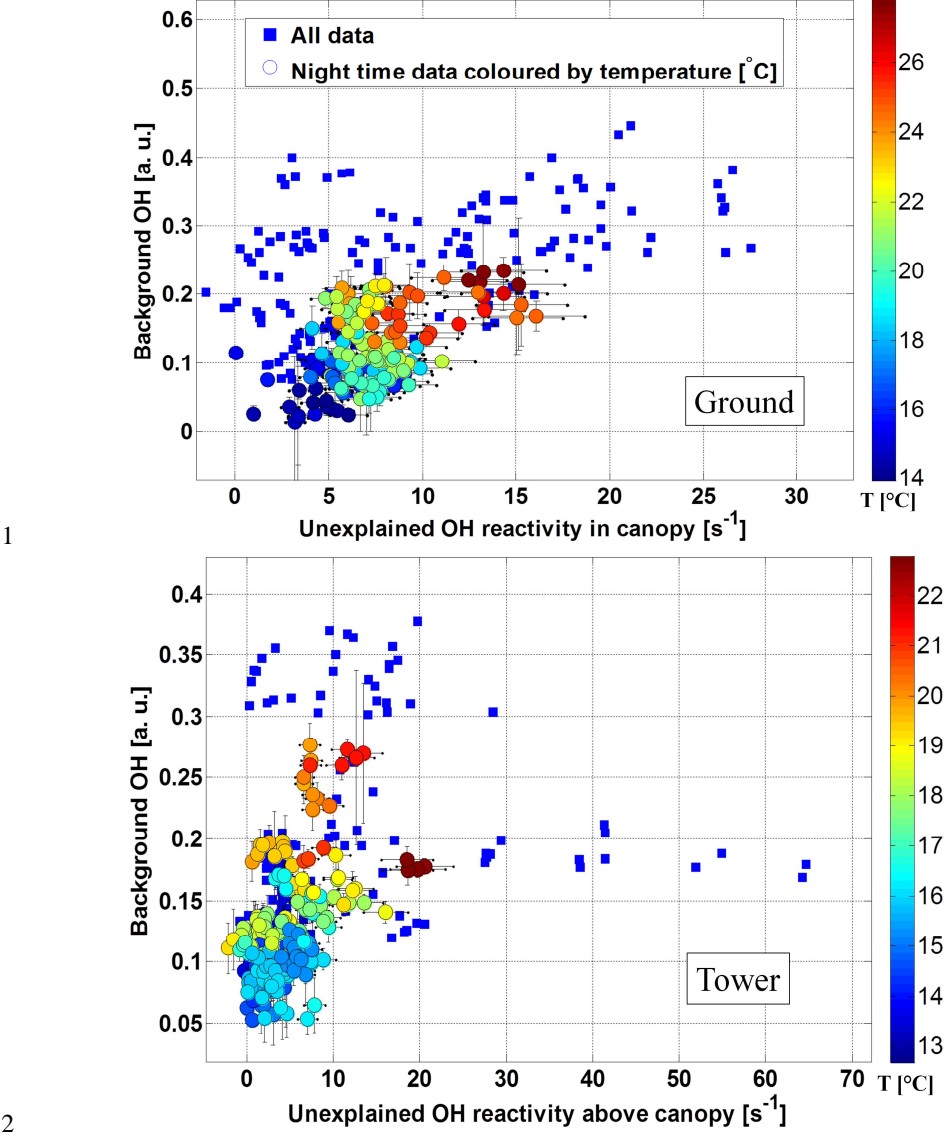

Figure 6. Background OH as a function of unexplained OH reactivity for ground and tower period measurements during the HUMPPA-COPEC 2010 campaign. Squares represent the daytime data, bullets represent night time data and are coloured accordingly to temperature (right legend).





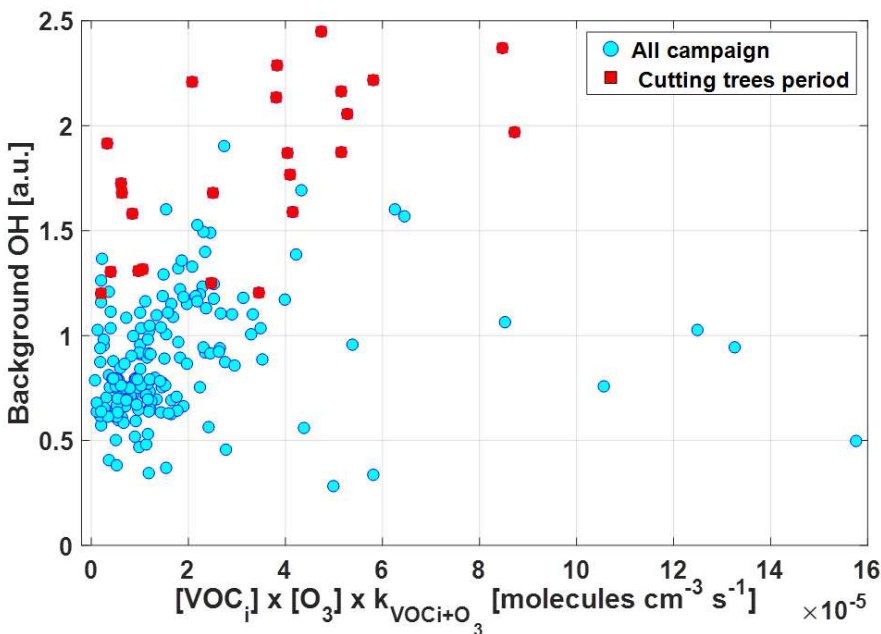

Figure 7. Background OH as a function of the sum of the product of the measured
unsaturated VOC-ozone turn-over (Table SI-1), during the HOPE 2012 campaign. The blue
points refers to the entire field campaign excluding the period of cutting trees, which is the
interval between 1st and 3rd of August 2012, described by the red squares.





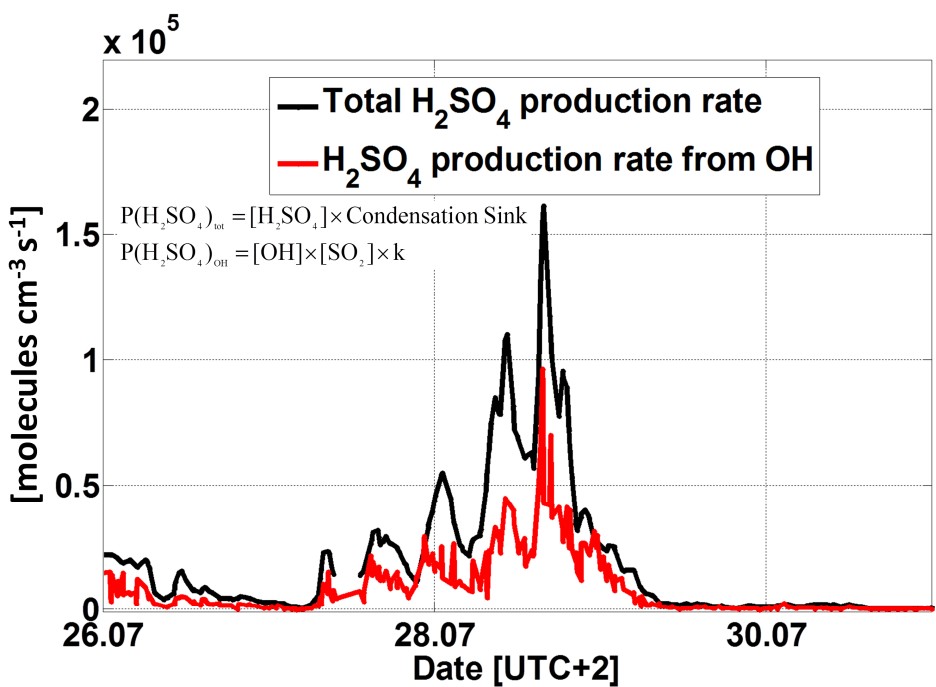

Figure 8. Comparison of the total H$_2$SO$_4$ production rate (black line), calculated from the
measured H$_2$SO$_4$, and the production rate of H$_2$SO$_4$ (red line) involving only the oxidation
process of SO$_2$ by OH for the ground measurements during the HUMPPA-COPEC 2010
campaign.





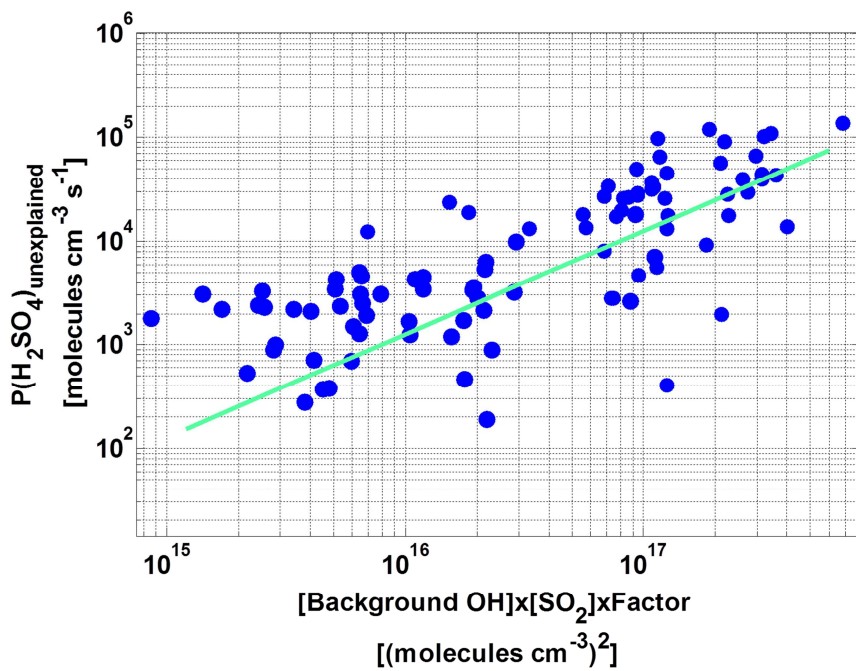

2    Figure 9. The production rate of $H_2SO_4$ unaccounted for by the oxidation of $SO_2$ by OH as a

3    function of the $OH_{bg}$ multiplied by $SO_2$ concentration during the ground measurements of the

4    HUMPPA-COPEC 2010 campaign. $OH_{bg}$ is expressed in molecules $cm^{-3}$ equivalents of OH.



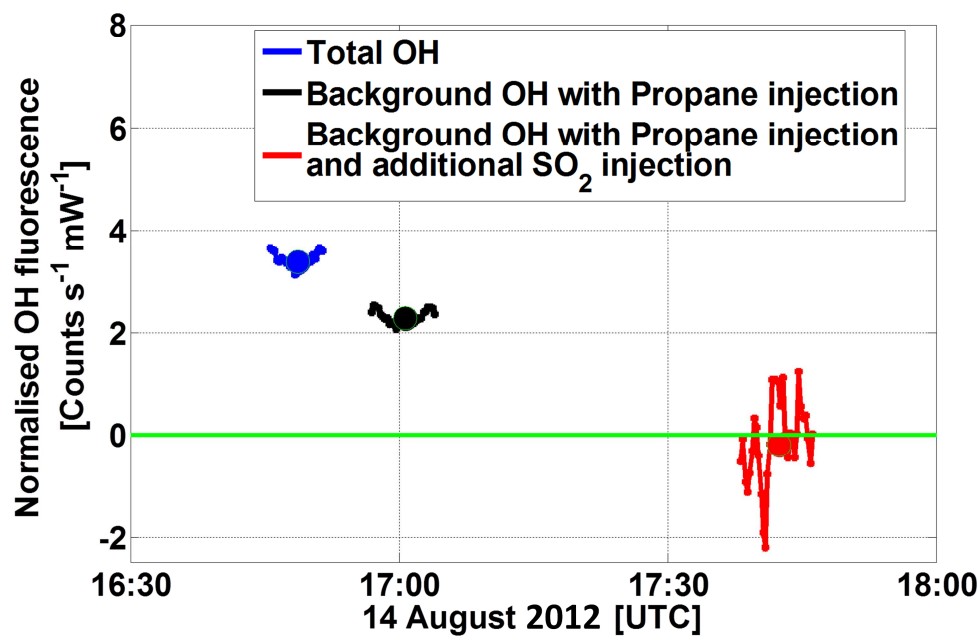

Figure 10. SO$_2$ injection test within IPI during the HOPE 2012 campaign. The blue data
points represent the total OH measured when no injection is performed. The black data points
represent the background OH when propane (2.5 x 10$^{15}$ molecules cm$^{-3}$) is scavenging > 90
% of ambient OH. The red signal is the background OH observed when in addition to
propane SO$_2$ (1 x 10$^{13}$ molecules cm$^{-3}$) is also injected.





Traditional nozzle    Molecular beam skimmer nozzle

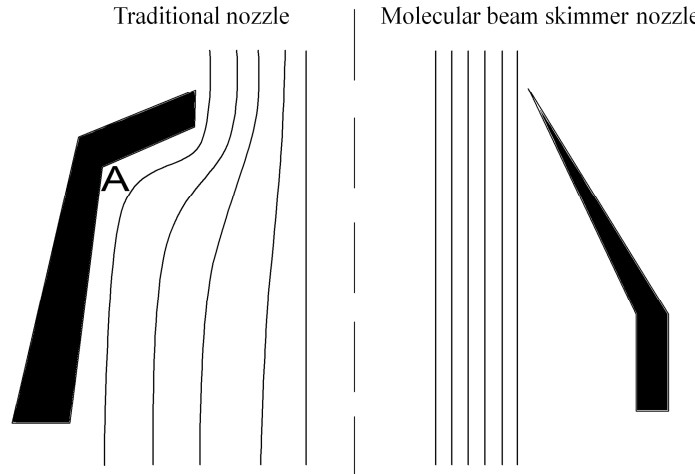

2    Figure 11. Schematic representation of the flow patterns in the traditional (left) and molecular

3    beam (right) nozzles. "A" indicates the area where deposition of particles is observed.