# Peer review of "Estimating the atmospheric concentration of Criegee intermediates and their possible interference in a FAGE-LIF"

_Atmospheric Chemistry and Physics, 2016_

## Referee Comment (RC1) · Anonymous Referee #1 · 1 Dec 2016

**Review of "Identifying Criegee Intermediates as Potential Oxidants in the Atmosphere" by A. Novelli, K. Hens, C. T. Ernest, M. Martinez, A. C. Nölscher, V. Sinha, P. Paasonen, T. Petäjä, M. Sipila, T. Elste, C. Plass-Dülmer, G. J. Phillips, D. Kubistin, J. Williams, L. Vereecken, J. Lelieveld, and H. Harder**

This paper focuses on the analysis of data from two field measurement campaigns, and combines results from laboratory experiments to assess relationships between inferred amounts of stabilized Criegee intermediates (SCI) in the atmosphere, and their possible contribution to the background signals in the measurement of hydroxyl radicals (OH) by a particular laser induced fluorescence (LIF)-based instrument. The physical chemistry of SCI is currently a subject of great interest, as is improved understanding of the components of the background in LIF-based OH instrumentation. Analysis of data collected during comprehensive ambient field campaigns is a sensible way to attack this problem. The authors have done a good job of precisely describing their assumptions and the potential pitfalls of various approaches. They have used various statistical parameters (e.g. correlation coefficients) to allow the reader to understand the robustness of the relationship under discussion. The literature is well-cited, and indeed the reference list serves as a useful list of recent Criegee intermediate-related papers.

**General comments.**

1)  This reviewer finds the discussion of the average concentration of SCI over an entire field campaign not very useful. It would be much better to work with an entire time series of SCI, followed by development of average diurnal cycles, and dependence of the derived concentrations on various parameters such as ozone, NOx, OH reactivity, reactivity from all BVOC, reactivity from all alkenes, and so on. This would give the reader a better feel as to what to expect from SCI behavior in various atmospheric environments.

2)  Regarding the rate coefficient for SCI with $SO_2$, this reviewer suggests going with the larger value obtained by several groups, and simply mentioning in the discussion that if the Mauldin et al., 2012 value is correct, then the concentrations of SCI increase by a factor of 66 (3.3E-11 / 5E-13). As the current discussion implies, the values derived with the Mauldin et al. value seem high when various constraints are imposed.

3)  The data for both studies is lacking measurements of important species. For example, HUMPPA-COPEC 2010 is missing some terpenes, mono-alkenes, small alkanes, and aromatic compounds. HOPE 2012 is more complete, but is missing hydrogen peroxide, HONO, and a pentane isomer. Both studies are light on oxygenated species (5 compounds). Other measurements such as $NO_y$, alkyl and peroxy nitrates, nitric acid, and $HO_x$ radicals would help to put the measurements in an overall context of pollutant level and oxidizing capacity. It should be possible to use other studies in the region to estimate concentrations of these other species, using perhaps, ratios to CO or some other approach.

4)  The presentation of data in a table with average values give the reader only one dimension of their behavior with time, radiation, temperature, and other species. Time series plots of all the species in the Supplemental Information would be preferred, but in their place, additional information in the table would be helpful. Suggest including with the average (mean) values, include median, and standard deviation of the mean. For species with significant diurnal variations, average diurnal plots would be useful. Also, values in mixing ratio units (e.g. ppbv and pptv) should be used rather than molecular units. Plots of average ozone, OH, and $HO_2$ versus NO or $NO_x$ would help the reader understand the chemical regime of the air masses measured. Such plots may help the authors decide how to divide the data into various measurement regimes (e.g. background, polluted, mixed, anthropogenic, biogenic).

5) In several places, the sum of the VOC concentration is given.  This reviewer is not convinced that this is a very useful parameter, since the various VOCs measured have very different reactivities with OH, ozone, and $NO_3$.  It might be better to use the sum of the OH reactivity due to the VOCs to describe the varying composition of air masses (e.g. when discussing the tree cutting event).

6) With a more complete set of species measured and estimated, it makes sense to perform model runs with the Master Chemical Mechanism (or other mechanism) modified as needed based on the latest information on SCI reactions and photolysis.  These runs will aid in the budget analysis of individual SCI and their sum, as well as budgets of other species (e.g. OH, sulfuric acid).

7) Some species and parameters are so important to the analyses described in this paper as to deserve further discussion on their validity (e.g. OH reactivity).  Have the methods been compared in the laboratory and/or the field with other techniques?  Do the values behave as expected and as seen in other studies as functions of time of day, $NO_x$, OH reactivity, etc?  While everyone in such campaigns submits their data in the best possible state that has been carefully quality assured and quality controlled, mistakes can be made.  Detailed data examinations can sometimes reveal such problems.

8) This reviewer is not convinced that the current title describes the content of the paper.  Suggest changing the title to more accurately reflect the paper contents and conclusions.

**Specific comments.**

Page 2, line 11.  Suggest "…through the addition to sulphur dioxide."

Page 2, line.11.  The term "central value" is used here and elsewhere in the paper.  Does this refer to the mean or median value or something else?  Suggest using standard statistical terms.

Page 4, line 6.  Suggest "…while lower limits have been determined…"

Page 5, line 23.  In giving an OH value, is this meant to be globally, daily maximum, or campaign average.  This value seems a bit high for the daily maximum remote locations, and perhaps a bit low for polluted situations.  Suggest adding a bit more text to describe what is mean by this value.

Page 6, line 1.  Suggest another word in place of "relevant", perhaps "significant".

Page 7, line 15.  An OH scavenger is mentioned, but its identify is not revealed.  Suggest saying specifically what compound is used.

Page 9, line 1.  Suggest "…titration of $OH_{atm}$ is performed…" and "…to determine their backgrounds in different environments."

Page 9, line 5.  The description of "better operation" is vague.  Suggest a bit more information or rewording.

Page 9, line 20.  Suggest "…Scots Pines…"

Page 9, line 22.  Suggest "…aerosol particle concentrations, size distributions…"

Page 10, line 6.  Check to make sure BVOC is defined earlier in the paper.

Page 10, line 6-8.  Given the importance of the OH reactivity measurements, suggest adding more information including the estimated uncertainties.

Page 11, line 22-25.  The discussion of this method of OH reactivity has a bit more information.  Suggest making the discussion similar for both methods.  Discussion of a comparison of these two techniques would be beneficial, if it has been done.

Page 12, line 14.  This equation arises out of the steady state assumption for sulfuric acid.  Suggest adding a bit of discussion of the assumptions that why steady state should be expected.

Page 13, line 1.  Suggest "…the rate coefficients of SCI…"

Page 13, line 2.  It is stated that the steady state concentration of SCI can be calculated, but equation (2) is that assuming steady state for sulfuric acid.  One can calculate the SCI value, but it is not the steady

state value. As discussed earlier, suggest doing this calculation for every data point for which all the data needed are available. Then perform statistics and comparison with the full time series.

Page 15, line 9-11. It is not clear what the steady state concentration (although it is not the steady state concentration, as discussed earlier) remains in agreement with. Suggest a bit more text to clarify this. Also, suggest "…based on the measured concentrations of VOC…"

Page 15, line 14. It appears to this reader that the calculation yields total sulfuric acid loss (rather than production). Of course, in steady state, production and loss are equal.

Page 16, equation 3. This equation yields values of individual SCI using the assumption of steady state. In addition to loss of SCI from unimolecular decomposition, It appears that reaction with water, and other reactions should be included. Also the equation should be clearer as to which terms are being summed. This reviewer believes that the summation should be over $k$, [VOC], $Y_{SCI}$, and the loss terms. Ozone is the only parameter that should be outside the summation. Suggest constructing a full time series for this estimate of SCI.

Page 16, line 8. Suggest "We assume…"

Page 16, line 13. Doesn't $Y_{SCI}$ depend on the VOC that reacts? If so, then the value should be different for each one.

Page 17, line 12. Suggest "…in a pristine forest environment…"

Page 18, line 24. Should the statement "…half of the measured OH…" actually be "…one-third of the measured OH…"?

Page 19, line 22. The phenomenon of unexplained OH production when OH reactivity is high is interesting. It makes sense to this reviewer that this observation could be due, at least in part, to errors in the measurement of OH reactivity. It also could show up when NOx is high, if NOx correlates with OH reactivity. If so, could it be to errors in measuring HONO or errors in the estimate of J(HONO)? The point is that it is not necessarily related to VOC levels or chemistry. Does the need for unexplained OH production disappear when the OH reactivity calculated from individual measured species is used rather than the directly measured OH reactivity?

Page 20, line 1. Suggest "…during low sun…" since a positive value for J(O1D), even though small, is not night time.

Page 20, line 15. The statement that only certain VOC are considered is confusing, because those VOC causing the unknown OH reactivity are themselves unknown. Suggest rewording this discussion.

Page 21, line 11. It is not clear exactly what is meant that the ratio of syn- to anti-SCI is within a factor of 5. Does that mean the ratio can go from 0.2 to 5? Or is anti-SCI always greater than or equal to syn-SCI? A bit more discussion would be helpful.

Page 22, line 18. Suggest "…which can represent an important primary source…" since whether HONO photolysis is important depends on conditions.

Page 23, line 1-2. You could assume a range of HONO levels and do the calculations with those to see what the impact is.

Page 23, section 3.5. While Figure 3 is a clever and useful way to show all the estimates of SCI concentrations, it might also be helpful to put the values into a table. Estimates of uncertainties for each value should also be given.

Page 24. Perhaps calculations with a range of unimolecular decomposition and other reactions would be instructive. Estimates on individual SCI species would be preferred. A discussion of the atmospheric impacts (e.g. oxidation of $SO_2$, oxidation of VOCs) of SCI over the range of values calculated would be welcome.

Page 25. While the sensitivity of the LIF instrument to OH cannot be used to calculate SCI concentrations from the OH background, this reviewer believes that the OH backgrounds should be normalized by the instrument sensitivity to account for internal instrument changes (e.g. laser power, white cell alignment) when discussing the OH background dependence on temperature and other parameters.

Page 25, line 10. Suggest saying that there is a strong correlation except for the 26-28 July (which are discussed later).

Page 25, line 21. This reviewer does not see how southern Germany has a larger fraction of non-biogenic VOC than Finland, based on the values in table SI-2. Suggest checking this statement for accuracy.

Page 29, line 11. Suggest "The reason for the lack of correlation…"

Page 31. In order to understand the possible impact of SCI on the instrument background, some information on residence times in various parts of the instrument would be helpful.

Page 32, line 6-8. Since OH produced by ozonolysis reactions quickly reacts, a mechanism for conversion of peroxy radicals to OH is critical. One possibility is $HO_2 + O_3$, and while not very fast, could be important.

Page 36, line 4. Suggest "Hence, higher SCI values appear to be less likely."

Page 38-48. Some of the references are not in alphabetical order.

Table 1. See earlier comments about Table SI-1 and 2. The "a" footnote is missing. Suggest adding more species (e.g. NOx, CO, etc.) to this table.

Figure 3. It appears the box for $k_{SCI+SO2} \sim 10^{-11}$ for HOPE 2012 should be moved to the left.

Figure 11. The flow lines for the skimmer do not seem reasonable. They should move toward the wall at least somewhat.

---

## Referee Comment (RC2) · Anonymous Referee #2 · 14 Dec 2016

This paper summaries steady-state calculations of the expected concentration of stabilized Criegee intermediates (SCIs) during two field campaigns (HUMPPA-COPEC 2010 and HOPE 2012). Several methods are used to estimate the concentration of SCIs in these environments, including estimates based on the missing $H_2SO_4$ oxidant, the ozonolysis of measured unsaturated compounds, unexplained total OH reactivity, and unexplained OH production rates. These different methods result in estimated SCI concentrations between 5 x $10^3$ and 2 x $10^6$ cm$^{-3}$ in these environments, although given the uncertainty associated with some of the assumptions used in these calculations the authors conclude that a value of 5 x $10^4$ cm$^{-3}$ with an uncertainty of approximately and order of magnitude is the most appropriate estimate of the SCI concentration in these environments.

The authors then provide empirical evidence that the artifact in their LIF-FAGE measurements during these field campaigns is the result of decomposition of SCIs in their low pressure detection cell. The evidence includes strong correlations of the observe OH background signal with temperature, ozone, and BVOC concentrations. In addition, scavenging experiments where $SO^2$ is added externally also removes the interference. However, the observed background OH signal corresponds to an equivalent concentration that is several orders of magnitude greater than the calculated SCI concentration suggesting that SCIs are not the only contributor to the background signal. Although the authors attempt to provide some possible explanations to account for this discrepancy, including a greater decomposition rate inside their detection cell and a different transmission efficiency of SCIs through their inlet compared to OH, they cannot fully explain the observed discrepancy.

The paper is well written and suitable for publication in ACP after the authors have addressed the following comments:

1) The title of the paper is somewhat misleading, as the paper does not explicitly identify Criegee intermediates given that the background OH signal cannot be solely attributed to SCIs. A more appropriate title might be "Estimating the concentration of Criegee intermediates as potential oxidants in the atmosphere."

2) The description of the different methods used to calculate the steady-state concentration is long and may detract from the overall conclusions of the paper. Moving some of this discussion to the Supplement would help maintain the focus of the paper on the resulting concentration estimates and interference discussion. Did the authors compare their estimations of the concentration of SCIs to that predicted by the Master Chemical Mechanism?

3) The strongest piece of evidence that the source of the OH background signal is due to SCIs is the $SO_2$ scavenging experiment described on page 30 and Figure 10. However, the paper would benefit from an expanded discussion of these measurements. What are the equivalent OH concentrations corresponding to the signals shown in Figure 10? Do they correspond to the high equivalent OH concentrations discussed on page 33? Did the authors attempt more than one scavenging experiment at different times during the day and/or night? Was the background signal consistently scavenged during multiple experiments? Were there periods when addition of $SO_2$ did not scavenge all of the background signal? Providing more details on these experiments would give additional confidence that SCIs were responsible for the high background OH signal.

---

## Author Response (AR1)

We thank the anonymous referees for reading the paper carefully and providing thoughtful comments, which have resulted in improvements in the revised version of the manuscript. We reply to each comment below in bold text.

Anonymous Referee #1

This paper focuses on the analysis of data from two field measurement campaigns, and combines results from laboratory experiments to assess relationships between inferred amounts of stabilized Criegee intermediates (SCI) in the atmosphere, and their possible contribution to the background signals in the measurement of hydroxyl radicals (OH) by a particular laser induced fluorescence (LIF)-based instrument.

The physical chemistry of SCI is currently a subject of great interest, as is improved understanding of the components of the background in LIF-based OH instrumentation. Analysis of data collected during comprehensive ambient field campaigns is a sensible way to attack this problem. The authors have done a good job of precisely describing their assumptions and the potential pitfalls of various approaches. They have used various statistical parameters (e.g. correlation coefficients) to allow the reader to understand the robustness of the relationship under discussion. The literature is well-cited, and indeed the reference list serves as a useful list of recent Criegee intermediate-related papers.

**General comments.**
1) This reviewer finds the discussion of the average concentration of SCI over an entire field campaign not very useful. It would be much better to work with an entire time series of SCI, followed by development of average diurnal cycles, and dependence of the derived concentrations on various parameters such as ozone, NOx, OH reactivity, reactivity from all BVOC, reactivity from all alkenes, and so on. This would give the reader a better feel as to what to expect from SCI behavior in various atmospheric environments.

**The authors understand the reviewers' concern, and a clarification is needed. The average value for the SCI concentrations from the different approaches in the two different environments is obtained from a time series of SCI. The time series are now shown in the supplementary information. Average diurnal cycles of the OH background signals from both campaigns discussed in this paper have been previously published by Novelli et al. (2014). The only exception is represented by the SCI estimate from the unexplained production rate of OH where the average value is used instead as it is directly taken from Hens et al. (2014). This is now better explained in the revised version of the manuscript. The authors do not support the suggestion of showing the dependence of the SCI concentration on various parameters; this is beyond the scope of this manuscript as its goal is to try and provide the boundary concentration of SCI in two environments rather than targeting the behavior of the SCI in general. The title of the manuscript has been changed in the revised version to better reflect this focus.**

2) Regarding the rate coefficient for SCI with $SO_2$, this reviewer suggests going with the larger value obtained by several groups, and simply mentioning in the discussion that if the Mauldin et al., 2012 value is correct, then the concentrations of SCI increase by a factor of 66 (3.3E-11 / 5E-13). As the current discussion implies, the values derived with the Mauldin et al. value seem high when various constraints are imposed.

**The authors would still prefer to keep the lower rate coefficient in the calculation of the SCI concentration from the sulfuric acid budget as this value is published and no different**

**measurement from the same group showing a different value is available yet. We have moved all of the discussion on possible causes of the difference in the rate coefficients to the supporting information (following the suggestion of reviewer #2) where we now also discuss a much broader range of possible sources of differences.**

3) The data for both studies is lacking measurements of important species. For example, HUMPPACOPEC 2010 is missing some terpenes, mono-alkenes, small alkanes, and aromatic compounds. HOPE 2012 is more complete, but is missing hydrogen peroxide, HONO, and a pentane isomer. Both studies are light on oxygenated species (5 compounds). Other measurements such as $NO_y$, alkyl and peroxy nitrates, nitric acid, and $HO_x$ radicals would help to put the measurements in an overall context of pollutant level and oxidizing capacity. It should be possible to use other studies in the region to estimate concentrations of these other species, using perhaps, ratios to CO or some other approach.

**The authors agree with the reviewer about the incompleteness of the data set for the HUMPPA-COPEC 2010 campaign, though we feel that the data set from HOPE 2012 is very extensive and covers a sufficient suite of trace gases. It seems unlikely, especially given the large uncertainties related directly to SCI yields, unimolecular decomposition rates and rate coefficients with other trace gases, that the addition of other trace gases would improve the estimates. Also, several of the compounds measured (alkanes, aromatics, HONO, $H_2O_2$) are not sources of SCI. As discussed in the answer to the first comment, the aim of the paper is to give boundaries for the SCI concentration, not to assess their role in the atmosphere.**

4) The presentation of data in a table with average values give the reader only one dimension of their behavior with time, radiation, temperature, and other species. Time series plots of all the species in the Supplemental Information would be preferred, but in their place, additional information in the table would be helpful. Suggest including with the average (mean) values, include median, and standard deviation of the mean. For species with significant diurnal variations, average diurnal plots would be useful. Also, values in mixing ratio units (e.g. ppbv and pptv) should be used rather than molecular units. Plots of average ozone, OH, and HO2 versus NO or NOx would help the reader understand the chemical regime of the air masses measured. Such plots may help the authors decide how to divide the data into various measurement regimes (e.g. background, polluted, mixed, anthropogenic, biogenic).

**For the time series of all the species from the HUMPPA-COPEC 2010 campaign we now refer directly to Hens et al. (2014) and Nölscher et al. (2012) which contain such data. For the HOPE 2012 campaign time series of the species listed in the tables have been added to the supplementary information. The authors did look into the possible division of the data into various chemical regimes but this did not produce any particular additional insight.**

5) In several places, the sum of the VOC concentration is given. This reviewer is not convinced that this is a very useful parameter, since the various VOCs measured have very different reactivities with OH, ozone, and $NO_3$. It might be better to use the sum of the OH reactivity due to the VOCs to describe the varying composition of air masses (e.g. when discussing the tree cutting event).

**We agree with the reviewer about the different reactivities VOCs have towards oxidants. Within this study we tried several times to underline this issue. For example, for the unexplained OH reactivity estimate of the SCI concentration (Chapter 3.3), we perform a sensitivity study (now moved to the supplementary information following the suggestion of reviewer #2) to account for the**

**different reactivity VOCs have towards OH and O$_3$. In the case of the estimate of the SCI concentration from the unexplained OH production (Chapter 3.4) we actually estimate the VOC ozone reactivity exactly to avoid the use of the sum of VOCs. This is better explained in the revised version of the manuscript. The delineation between the tree cutting period and the rest of the "normal" campaign is based on the differences in the measured and unexplained OH reactivity due to the VOCs.**

6) With a more complete set of species measured and estimated, it makes sense to perform model runs with the Master Chemical Mechanism (or other mechanism) modified as needed based on the latest information on SCI reactions and photolysis. These runs will aid in the budget analysis of individual SCI and their sum, as well as budgets of other species (e.g. OH, sulfuric acid).

**The authors agree that modelling with the MCM can be beneficial, but for this particular paper we consider it to be out of scope, as it is very long already. Currently, the model would not bring any additional meaningful information on the budget beyond what has been already done in the manuscript, given the uncertainties associated with the chemistry of SCI. Moreover, we currently do not have the necessary information to speciate the SCI (we have rate coefficients for only a handful of them) and a model study would not provide this.**

7) Some species and parameters are so important to the analyses described in this paper as to deserve further discussion on their validity (e.g. OH reactivity). Have the methods been compared in the laboratory and/or the field with other techniques? Do the values behave as expected and as seen in other studies as functions of time of day, NO$_x$, OH reactivity, etc? While everyone in such campaigns submits their data in the best possible state that has been carefully quality assured and quality controlled, mistakes can be made. Detailed data examinations can sometimes reveal such problems.

**The authors understand and share the concerns of the reviewer regarding the quality and validity of the data used in this or any study. We have no reason to doubt the data provided by our fellow colleagues, who have checked their validity. Furthermore, most of the data has been already published in other manuscripts and was critically evaluated at that time. For other parameters we have included extensive discussions as to the expected uncertainties (e.g. k(T) of SCI + SO$_2$, condensation H$_2$SO$_4$ sink,...).**

8) This reviewer is not convinced that the current title describes the content of the paper. Suggest changing the title to more accurately reflect the paper contents and conclusions.

**The authors agree with the reviewer and, also in response to the comments of reviewer#2, changed the paper title into: "Estimating the atmopsheric concentration of Criegee intermediates and their possible interference in a FAGE-LIF instrument".**

**Specific comments.**
Page 2, line 11. Suggest "…through the addition to sulphur dioxide."

**The sentence was rephrased.**

Page 2, line.11. The term "central value" is used here and elsewhere in the paper. Does this refer to the mean or median value or something else? Suggest using standard statistical terms.

**The term central value was replaced with the term average.**

Page 4, line 6. Suggest "…while lower limits have been determined…"

**Changed accordingly.**

Page 5, line 23. In giving an OH value, is this meant to be globally, daily maximum, or campaign average.
This value seems a bit high for the daily maximum remote locations, and perhaps a bit low for polluted situations. Suggest adding a bit more text to describe what is mean by this value.

**The value of OH was meant to give an idea of the order of magnitude of the OH concentration globally, but as it was misleading it was removed from the manuscript.**

Page 6, line 1. Suggest another word in place of "relevant", perhaps "significant".

**Changed accordingly.**

Page 7, line 15. An OH scavenger is mentioned, but its identify is not revealed. Suggest saying specifically what compound is used.

**Changed accordingly.**

Page 9, line 1. Suggest "…titration of $OH_{atm}$ is performed…" and "…to determine their backgrounds in different environments."

**The sentence was unclear and was rephrased.**

Page 9, line 5. The description of "better operation" is vague. Suggest a bit more information or rewording.

**Changed accordingly.**

Page 9, line 20. Suggest "…Scots Pines…"

**Changed accordingly.**

Page 9, line 22. Suggest "…aerosol particle concentrations, size distributions…"

**Changed accordingly.**

Page 10, line 6. Check to make sure BVOC is defined earlier in the paper.

**The definition of BVOC has been added.**

Page 10, line 6-8. Given the importance of the OH reactivity measurements, suggest adding more information including the estimated uncertainties.

**A short description of the technique together with the uncertainty on the measurement was added.**

Page 11, line 22-25. The discussion of this method of OH reactivity has a bit more information. Suggest making the discussion similar for both methods. Discussion of a comparison of these two techniques would be beneficial, if it has been done.

**There is no peer-reviewed information available comprising an intercomparison between these two techniques.**

Page 12, line 14. This equation arises out of the steady state assumption for sulfuric acid. Suggest adding a bit of discussion of the assumptions that why steady state should be expected.

**A discussion on the validity of the assumption of steady state for sulfuric acid was added.**

Page 13, line 1. Suggest "…the rate coefficients of SCI…"

**Changed accordingly.**

Page 13, line 2. It is stated that the steady state concentration of SCI can be calculated, but equation (2) is that assuming steady state for sulfuric acid. One can calculate the SCI value, but it is not the steady state value. As discussed earlier, suggest doing this calculation for every data point for which all the data needed are available. Then perform statistics and comparison with the full time series.

**Please refer to the answer given to the first general comment.**

Page 15, line 9-11. It is not clear what the steady state concentration (although it is not the steady state concentration, as discussed earlier) remains in agreement with. Suggest a bit more text to clarify this. Also, suggest "…based on the measured concentrations of VOC…"

**The sentence has been rephrased.**

Page 15, line 14. It appears to this reader that the calculation yields total sulfuric acid loss (rather than production). Of course, in steady state, production and loss are equal.

**While we agree with the reviewer, as discussed in the comments above, we think that the assumption of steady state is reasonable within the uncertainty of our study and therefore production and loss rate are equal.**

Page 16, equation 3. This equation yields values of individual SCI using the assumption of steady state. In addition to loss of SCI from unimolecular decomposition, It appears that reaction with water, and other reactions should be included. Also the equation should be clearer as to which terms are being summed. This reviewer believes that the summation should be over k, [VOC], $Y_{SCI}$, and the loss terms. Ozone is the only parameter that should be outside the summation. Suggest constructing a full time series for this estimate of SCI.

**The loss of SCI of 40 s$^{-1}$ already includes reaction with water, water dimers and several trace gases as described in the text as this value is obtained from a previous work. (Novelli et al., 2014b). The formula has been changed as requested. Due to the limited information available, our analysis uses the same $Y_{SCI}$ and loss processes as an average value for all SCI, such that they can be factored out of the summation, which is what we did in the original version.**

Page 16, line 8. Suggest "We assume…"

**Changed accordingly.**

Page 16, line 13. Doesn't $Y_{SCI}$ depend on the VOC that reacts? If so, then the value should be different for each one.

**See answer to previous comment.**

Page 17, line 12. Suggest "…in a pristine forest environment…"

**Changed accordingly.**

Page 18, line 24. Should the statement "…half of the measured OH…" actually be "…one-third of the measured OH…"?

**The sentence was unclear and has been rephrased.**

Page 19, line 22. The phenomenon of unexplained OH production when OH reactivity is high is interesting.
It makes sense to this reviewer that this observation could be due, at least in part, to errors in the measurement of OH reactivity. It also could show up when NOx is high, if NOx correlates with OH reactivity. If so, could it be to errors in measuring HONO or errors in the estimate of J(HONO)? The point is that it is not necessarily related to VOC levels or chemistry. Does the need for unexplained OH production disappear when the OH reactivity calculated from individual measured species is used rather than the directly measured OH reactivity?

**In the Hens et al. (2014) study from which the unexplained OH production is taken there is a detailed analysis that takes into account how the uncertainty on individual data would impact the OH budget. The conclusion was that an unexplained OH production would still appear for large OH reactivity. The OH reactivity calculated from only the measured species gives a value of $2 \pm 2$ s$^{-1}$ for the entire duration of the campaign. If such a value is used for the OH budget, the production rate would exceed the loss rate by a factor of 4 for any conditions. Therefore, we believe that the measured total OH reactivity better represents the chemistry in the forest.**

Page 20, line 1. Suggest "…during low sun…" since a positive value for J(O1D), even though small, is not night time.

**The authors would prefer to keep the text as is as the distinction between the two different ranges of J(O$^1$D) was taken from Hens et al. (2014) where it was necessary to choose a value of J(O$^1$D) to separate between daylight conditions (photolysis of trace gases) and night conditions. We agree with the reviewer that a small positive value of J(O$^1$D) indicates presence of light but we find the simplification of defining such low values for night-time periods suitable for the paper and easier to follow for the reader.**

Page 20, line 15. The statement that only certain VOC are considered is confusing, because those VOC causing the unknown OH reactivity are themselves unknown. Suggest rewording this discussion.

**The discussion was reworded.**

Page 21, line 11. It is not clear exactly what is meant that the ratio of syn- to anti-SCI is within a factor of 5. Does that mean the ratio can go from 0.2 to 5? Or is anti-SCI always greater than or equal to syn-SCI? A bit more discussion would be helpful.

**The paragraph was edited for clarity.**

Page 22, line 18. Suggest "…which can represent an important primary source…" since whether HONO photolysis is important depends on conditions.

**Changed accordingly.**

Page 23, line 1-2. You could assume a range of HONO levels and do the calculations with those to see what the impact is.

**The authors think that the SCI estimate that would be obtained by assuming a certain concentration of HONO would carry such a large uncertainty that the resulting SCI concentration would be meaningless.**

Page 23, section 3.5. While Figure 3 is a clever and useful way to show all the estimates of SCI concentrations, it might also be helpful to put the values into a table. Estimates of uncertainties for each value should also be given.

**A table of the estimated SCI concentrations was added.**

Page 24. Perhaps calculations with a range of unimolecular decomposition and other reactions would be instructive. Estimates on individual SCI species would be preferred. A discussion of the atmospheric impacts (e.g. oxidation of $SO_2$, oxidation of VOCs) of SCI over the range of values calculated would be welcome.

**The authors agree with the reviewer about the importance of estimates of individual SCI species and the benefit of discussing the atmospheric impact SCI would have in the two specific environments. Unfortunately, the current data available does not allow for a SCI specific estimate. For example, there is little to no information on the unimolecular decomposition rate for speciated SCI. An estimate of a minor impact that SCI would have, for example towards $SO_2$, is discussed. We consider a more in depth analysis of the SCI impact on several trace gases not within the scope of this study.**

Page 25. While the sensitivity of the LIF instrument to OH cannot be used to calculate SCI concentrations from the OH background, this reviewer believes that the OH backgrounds should be normalized by the instrument sensitivity to account for internal instrument changes (e.g. laser power, white cell alignment) when discussing the OH background dependence on temperature and other parameters.

**The sensitivity to OH is normalized by a calibration procedure described in the instrumental description (Chapter 2.1); the OH background is normalized to the laser power, possible change in the alignment of the white cell and quenching due to water vapor, all factors which would affect both atmospheric OH and background OH concentrations.**

Page 25, line 10. Suggest saying that there is a strong correlation except for the 26-28 July (which are discussed later).

**The authors think that there may be some confusion. The specific period of 26-28 July discussed later refers to the HOPE 2012 campaign while the reviewer points at page 25, line 10 when the**

**discussion is about the HUMPPA-COPEC 2010 campaign where we have no solid reasons to exclude the 26-28 July period.**

Page 25, line 21. This reviewer does not see how southern Germany has a larger fraction of non-biogenic VOC than Finland, based on the values in table SI-2. Suggest checking this statement for accuracy.

**The statement was misleading and it has been rephrased.**

Page 29, line 11. Suggest "The reason for the lack of correlation…"

**Changed accordingly.**

Page 31. In order to understand the possible impact of SCI on the instrument background, some information on residence times in various parts of the instrument would be helpful.

**Information about the residence time of the sampled air between the pinhole and the area where the OH is detected which is relevant for the decomposition of SCI in the LIF instrument was added.**

Page 32, line 6-8. Since OH produced by ozonolysis reactions quickly reacts, a mechanism for conversion of peroxy radicals to OH is critical. One possibility is $HO_2 + O_3$, and while not very fast, could be important.

**The reactions suggested by the reviewer were already included in the model study performed in Hens et al. (2014) and did not improve the agreement between measured and calculated OH radicals.**

Page 36, line 4. Suggest "Hence, higher SCI values appear to be less likely."

**Changed accordingly.**

Page 38-48. Some of the references are not in alphabetical order.

**Checked.**

Table 1. See earlier comments about Table SI-1 and 2. The "a" footnote is missing. Suggest adding more species (e.g. NOx, CO, etc.) to this table.

**Changed accordingly.**

Figure 3. It appears the box for $k_{SCI+SO_2} \sim 10^{-11}$ for HOPE 2012 should be moved to the left.

**Changed accordingly.**

Figure 11. The flow lines for the skimmer do not seem reasonable. They should move toward the wall at least somewhat.

**The schematic shown is very simplified. We agree with the reviewer, but the main purpose of the figure is to underline the large difference in how the air evolves in the low pressure region of the instrument depending on the typology of nozzle, not to reproduce the air pattern rigorously.**

**Nevertheless as the figure is not critical, and possibly misleading, it was removed from the manuscript.**

**Anonymous Referee #2**

This paper summaries steady-state calculations of the expected concentration of stabilized
Criegee intermediates (SCIs) during two field campaigns (HUMPPA-COPEC
and HOPE 2012). Several methods are used to estimate the concentration of
SCIs in these environments, including estimates based on the missing $H_2SO_4$ oxidant,
the ozonolysis of measured unsaturated compounds, unexplained total OH reactivity,
and unexplained OH production rates. These different methods result in estimated
SCI concentrations between $5 \times 10^3$ and $2 \times 10^6$ cm$^{-3}$ in these environments, although
given the uncertainty associated with some of the assumptions used in these
calculations the authors conclude that a value of $5 \times 10^4$ cm$^{-3}$ with an uncertainty
of approximately and order of magnitude is the most appropriate estimate of the SCI
concentration in these environments.
The authors then provide empirical evidence that the artifact in their LIF-FAGE mea-
surements during these field campaigns is the result of decomposition of SCIs in their
low pressure detection cell. The evidence includes strong correlations of the observe
OH background signal with temperature, ozone, and BVOC concentrations. In addition,
scavenging experiments where $SO_2$ is added externally also removes the interference.
However, the observed background OH signal corresponds to an equivalent concentration
that is several orders of magnitude greater than the calculated SCI concentration
suggesting that SCIs are not the only contributor to the background signal. Although
the authors attempt to provide some possible explanations to account for this discrepancy,
including a greater decomposition rate inside their detection cell and a different
transmission efficiency of SCIs through their inlet compared to OH, they cannot fully
explain the observed discrepancy.
The paper is well written and suitable for publication in ACP after the authors have
addressed the following comments:
1) The title of the paper is somewhat misleading, as the paper does not explicitly identify
Criegee intermediates given that the background OH signal cannot be solely attributed
to SCIs. A more appropriate title might be "Estimating the concentration of Criegee
intermediates as potential oxidants in the atmosphere."

**Please refer to the answer to the general comment 7 given to the reviewer #1. We propose
"Estimating the atmospheric concentration of Criegee intermediates and their possible interference
in a FAGE-LIF instrument".**

2) The description of the different methods used to calculate the steady-state concentration
is long and may detract from the overall conclusions of the paper. Moving
some of this discussion to the Supplement would help maintain the focus of the paper
on the resulting concentration estimates and interference discussion. Did the authors
compare their estimations of the concentration of SCIs to that predicted by the Master
Chemical Mechanism?

**Part of the discussion of the SCI estimates, in particular the discussion about uncertainty in the rate
coefficients used in this study, was moved to the supplementary information.**

**A comparison of the SCI estimates with the MCM was not performed as the current MCM mechanism still lacks the majority of reactions and SCI conformers that would be needed in order to compare the values, i.e. there is little to compare against. In order to use the MCM to calculate the SCI concentration from different monoterpenes a lot of changes to the model would be needed, which is out of the scope of this study, though it is part of an upcoming study.**

3) The strongest piece of evidence that the source of the OH background signal is due to SCIs is the SO2 scavenging experiment described on page 30 and Figure 10. However, the paper would benefit from an expanded discussion of these measurements. What are the equivalent OH concentrations corresponding to the signals shown in Figure 10? Do they correspond to the high equivalent OH concentrations discussed on page 33? Did the authors attempt more than one scavenging experiment at different times during the day and/or night? Was the background signal consistently scavenged during multiple experiments? Were there periods when addition of SO2 did not scavenge all of the background signal? Providing more details on these experiments would give additional confidence that SCIs were responsible for the high background OH signal.

**The discussion regarding the $SO_2$ scavenging experiments was somewhat expanded as suggested by the reviewer. In the following, please find some brief answers to the questions raised by the reviewer. The equivalent of the OH concentration corresponding to the signal shown in Figure 10 would be ~ 2 x $10^6$ molecules $cm^{-3}$ and it would match with the values described at page 33. During the HOPE campaign we performed a few experiments at the very end of the campaign during the day and the results were similar to what is shown in figure 10. Since we put forward the hypothesis that the OH background observed in the LIF could be caused by the unimolecular decomposition of SCI, scavenging experiments with $SO_2$ were performed for every subsequent field campaign. In every environment in which the instrument was deployed, it was possible to scavenge the OH background signal with $SO_2$. This data will be the subject of future publication.**

**Relevant changes in the manuscript**

- Title of the manuscript was changed to: "Estimating the atmospheric concentration of Criegee intermediates and their possible interference in a FAGE-LIF instrument.
- Time series of the SCI concentrations estimated and of all relevant trace gases for the HOPE 2012 campaign were added in the supplementary information.
- Analysis of uncertainties from the $H_2SO_4$ (Chapter 3.1) and OH reactivity (Chapter 3.3) estimates were moved to the supplementary information.
- The analysis on the reason for discrepancy in the SCI + $SO_2$ rate coefficient in the supplementary information was expanded.
- A table with the average values for the SCI concentration estimated was added.
- Figure 11 was removed.
- Several paragraphs were rephrased and modified for clarity.

[revised manuscript text omitted]

**Assessment of the available rate coefficients for the SCI + SO$_2$ reaction**

The disagreement between the rate coefficient for the SCI + SO$_2$ reaction obtained by

Mauldin III et al. (2012) and Berndt et al. (2012), $5 \times 10^{-13}$ cm$^3$ molecule$^{-1}$ s$^{-1}$, and the one obtained by a number of other groups (Welz et al., 2012;Taatjes et al., 2013;Liu et al.,

2014b;Sheps et al., 2014;Stone et al., 2014;Chhantyal-Pun et al., 2015;Newland et al.,

2015a;Newland et al., 2015b;Foreman et al., 2016;Zhu et al., 2016), $3.3 \times 10^{-11}$ cm$^3$ molecule$^{-}$

$^1$ s$^{-1}$, is not straightforward to explain.

A first factor is that Mauldin III et al. (2012) and Berndt et al. (2012) measure the rate of formation of H$_2$SO$_4$ rather than the loss of SCI by SO$_2$. Theoretical and experimental results (Carlsson et al., 2012;Ahrens et al., 2014) indicate that SO$_3$ is the main product of the SCI +

SO$_2$ reaction, with a yield near 100% at all reaction conditions considered. Barring secondary chemistry removing SO$_3$ prior to its reaction with H$_2$O to form H$_2$SO$_4$, which seems unlikely under their reaction conditions, the H$_2$O$_4$ yield should match the SCI loss. Earlier theoretical work by Vereecken et al. (2012) suggested that the secondary ozonide (SOZ) formed as an intermediate from the reaction between larger SCI and SO$_2$ could stabilize and undergo bimolecular reaction without formation of SO$_3$;  the loss of SOZ would then reduce SO$_3$

formation, explaining the difference in the rate coefficients for the different experiments.

However, more recent theoretical work (Kuwata et al., 2015) found additional low-lying pathways that make collisional stabilization of the SOZ unlikely. Experiments by Carlsson et al. (2012) and Ahrens et al. (2014) observed high yields of SO$_3$ close to unity suggesting that the SOZ is not lost under the conditions used, i.e. in chambers with high concentrations of reactants and in the absence of water.

A second factor is that the reaction conditions used by Mauldin III et al. (2012) and Berndt et al. (2014) differ from the other studies, i.e. they were performed either at ambient air conditions or with lower concentrations of reagents and in the presence of water, while the remaining experiments were typically performed under lower pressures, without efficient colliders present. The mechanism of the $SCI+SO_2$ reaction as obtained by several authors (Vereecken et al., 2012;Kuwata et al., 2015;Jiang et al., 2010;Kurtén et al., 2011) all indicate a barrierless formation of a pre-reactive complex or cycloadduct. This type of reactions typically show faster rate coefficients at higher pressures due to lower redissociation of the adduct; this is corroborated by the theoretical study on the pressure dependence by Kuwata et al. (2015) who finds no pressure dependence up to 10132.5 hPa and an increase in the effective rate coefficient for higher pressures. Experimental studies of the pressure dependence (Liu et al., 2014b;Huang et al., 2015;Chhantyal-Pun et al., 2016) do not show extensive pressure dependence up to 300 Torr, and all show a positive pressure dependence, in line with the currently accepted reaction mechanism. The Carlsson et al. (2012) experiments at 1013.51 hPa likewise can be fitted using a faster $CI + SO_2$ rate coefficient of 1 x $10^{-11}$ $cm^3$ molecule$^{-1}$ s$^{-1}$. This suggests that the reaction conditions used by Mauldin III et al. and Berndt et al. would likely lead to faster rate coefficients, especially for larger SCI as used in Mauldin III et al. due to the lower redissociation rate and hence higher thermalization yield of the intermediates.

A third factor is that the Mauldin III et al. examine SCI formed from larger terpenoids, rather than the smaller SCI examined in the remaining studies. A larger SCI should mean a longer lifetime for the SOZ intermediate, especially as this longer lifetime makes collisional thermalization more efficient. Hence the large SOZ might indeed live long enough to react in bimolecular reactions prior to dissociation to $SO_3$, contrary to smaller SOZ. Unfortunately, SOZ chemistry has not been studied in detail; for the current case the reaction with e.g. $H_2O$

could be a potential loss process. A prerequisite for this scavenging to be effective is that no

$H_2SO_4$ precursor should be formed. This scavenging of the SOZ intermediate would however not apply to the experiments of Berndt et al., which examined $CH_3CHOO$ and $(CH_3)_2COO$

Criegee intermediates, similar in size to those used in the studies yielding higher rate coefficients. For SCI of this size, the RRKM master equation analysis of (Kuwata et al., 2015)

predicts very fast SOZ decomposition.

Finally, an alternative explanation could be based on analysis of the studies by Mauldin III et al. (2012) and Berndt et al. (2012). In their experiments, the rate of the SCI+$SO_2$ reaction is derived relative to the total loss rate of SCI, $L_{SCI}$, as it governs the steady-state concentration of SCI with negligible $SO_2$ present. This $L_{SCI}$ has a value on the order of ~ 3 to 5 $s^{-1}$ in both experiments. Since these studies, a large body of experimental and theoretical data has become available, regarding the reactivity of SCI towards many coreactants present in the reaction mixture (Taatjes et al., 2013;Ouyang et al., 2013;Ahrens et al., 2014;Buras et al.,

2014;Liu et al., 2014a;Stone et al., 2014;Sheps et al., 2014;Welz et al., 2014;Lewis et al.,

2015). From this new data, we should consider that a total loss rate of about 4 $s^{-1}$ is an underestimate. In a previous study by Novelli et al. (2014) a value of $L_{SCI}$ = 40 $s^{-1}$ under atmospheric conditions was proposed. A re-analysis of the study by Mauldin III et al. (2012)

using $L_{SCI}$ = 40 $s^{-1}$ and the measured yield of SCI for α-pinene of 0.1 (Donahue et al., 2011), results in a rate coefficient for the α-pinene-derived SCI + $SO_2$ reaction of $2.6 \times 10^{-11}$ $cm^3$

molecule$^{-1}$ $s^{-1}$. Likewise, for the other compounds examined in the two studies (Berndt et al.,

2012;Mauldin III et al., 2012), the derived rate of SCI+$SO_2$ would shift significantly towards the higher values obtained in the other studies (Welz et al., 2012;Taatjes et al., 2013;Liu et al.,

2014b;Sheps et al., 2014;Stone et al., 2014). One must consider, though, that the study by

Berndt et al. (2012) included a measurement of $k_{loss}$, based on the observed $H_2SO_4$ formation from the steady state SCI in the absence of $SO_2$. Hence, this second explanation is only viable if another source of H₂SO₄ exists in the system; this has already been suggested by Newland if another source of $H_2SO_4$ exists in the system; this has already been suggested by Newland et al. (2015a) based on their $SO_2$ oxidation experiments.

Still, as these considerations for the lower values by Mauldin III et al. (2012) and Berndt et al.

(2012) are merely speculative, we will consider both $3.3 \times 10^{-11}$ cm$^3$ molecule$^{-1}$ s$^{-1}$ and $5 \times 10^{-13}$

cm$^3$ molecule$^{-1}$ s$^{-1}$ as possible rate coefficients for the SCI + $SO_2$ reaction in the current budget analysis.

**Sensitivity study on the unexplained OH reactivity SCI estimate**

The estimate of SCI from the unexplained OH reactivity data contains larger uncertainties compared to the previous estimates as the rate coefficient for ozonolysis of unsaturated compounds varies by up to three orders of magnitude. In addition, the rate coefficient between

OH and unsaturated compounds, depending on whether these are unsaturated NMHC or

OVOC, primary emissions, or secondary oxidation products, varies by an order of magnitude.

A sensitivity study was done on the SCI estimates from the unexplained OH reactivity to attempt to account for this uncertainty in rate coefficients. It is possible to calculate a lower limit for the SCI concentration by using the highest rate coefficient between OH and unsaturated compounds, $1 \times 10^{-10}$ cm$^3$ molecule$^{-1}$ s$^{-1}$ (Atkinson et al., 2006) combined with a slow rate coefficient for the unsaturated compounds and ozone, $1 \times 10^{-17}$ cm$^3$ molecule$^{-1}$ s$^{-1}$

(Atkinson et al., 2006), leading to a [SCI] = $(8.7 \pm 8.0) \times 10^3$ molecules cm$^{-3}$.  For the upper limit, a slower rate coefficient for OH and unsaturated OVOC, $\sim 3 \times 10^{-11}$ cm$^3$ molecule$^{-1}$ s$^{-1}$

(Atkinson et al., 2006;Teruel et al., 2006) together with a higher rate coefficient with $O_3$, $1 \times$

$10^{-16}$ cm$^3$ molecule$^{-1}$ s$^{-1}$ (Atkinson et al., 2006) results in a concentration of [SCI] = $(3 \pm 3) \times$

$10^5$ molecules cm$^{-3}$. These are the values obtained for the HUMPPA-COPEC 2010 campaign.

For the HOPE 2012 campaign, the same assumptions would yield a lower and an upper limit of $(1 \pm 0.2) \times 10^3$ molecules cm$^{-3}$ and $(2.9 \pm 0.7) \times 10^4$ molecules cm$^{-3}$, respectively.

Table SI-1. Average concentrations with 1σ standard deviation of measured unsaturated VOC

during the HUMPPA-COPEC 2010 and HOPE 2012 campaigns, together with the rate coefficients of the reaction with ozone (IUPAC recommended values) (Atkinson et al., 2006).

| Compound | [molecule cm$^{-3}$] | | Rate coefficient with O$_3$ [cm$^3$ molecule$^{-1}$ s$^{-1}$] |
| --- | --- | --- | --- |
| | HUMPPA-COPEC 2010 | HOPE 2012 | |
| isoprene | $(1.8 \pm 1.8)$ x $10^9$ | $(2.2 \pm 2.2)$ x $10^9$ | 1 x $10^{-14}$ exp(-1995/T) |
| α-pinene | $(2.7 \pm 3)$ x $10^9$ | $(1.5 \pm 1.5)$ x $10^9$ | 8.1 x $10^{-16}$ exp(-640/T) |
| β-pinene | $(1.9 \pm 6.6)$ x $10^8$ | $(9 \pm 9)$ x $10^8$ | 1.4 x $10^{-15}$ exp(-1270/T) |
| 3-carene | $(1.7 \pm 2)$ x $10^9$ | $(5.6 \pm 4.7)$ x $10^8$ | 4.8 x $10^{-17, b}$ |
| myrcene | $(2.6 \pm 2.7)$ x $10^8$ | $(2.2 \pm 1.6)$ x $10^8$ | 2.7 x $10^{-15}$ exp(-520/T) |
| limonene | n.a. | $(2.9 \pm 2.1)$ x $10^8$ | 2.8 x $10^{-15}$ exp(-770/T) |
| sabinene | n.a. | $(9.2 \pm 9.6)$ x $10^8$ | 8.2 x $10^{-17, b}$ |
| γ-terpinene | n.a. | $(1 \pm 1)$ x $10^8$ | 1.5 x $10^{-16, b}$ |
| 2-methylpropene | n.a. | $(4.2 \pm 2.5)$ x $10^8$ | 2.7 x $10^{-15}$ exp(-1630/T) |
| but-1-ene | n.a. | $(1.4 \pm 4.2)$ x $10^8$ | 1.2 x $10^{-17, a,b}$ |
| propene | n.a. | $(4.7 \pm 3.7)$ x $10^8$ | 5.5 x $10^{-15}$ exp(-1880/T) |
| cis-2-butene | n.a. | $(6.1 \pm 3.0)$ x $10^7$ | 3.2 x $10^{-15}$ exp(-965/T) |
| ethene | n.a. | $(7.3 \pm 9.0)$ x $10^9$ | 9.1 x $10^{-15}$ exp(-2580/T) |

a, rate coefficient from Adeniji et al. (1981).
b, at 298 K
1 ppbv = 2.46 x $10^{10}$ molecules cm$^{-3}$ at 295K and 1013 hPa.

Table SI-2. Average concentrations with 1σ standard deviation of measured trace gas during the HUMPPA-COPEC 2010 and HOPE 2012 campaigns, with the rate coefficients of the reaction with OH (IUPAC recommended values) (Atkinson et al., 2006;Atkinson et al., 2004)

| Compound | [molecule cm$^{-3}$] | | Rate coefficient with OH [cm$^3$ molecule$^{-1}$ s$^{-1}$] |
| --- | --- | --- | --- |
| | HUMPPA-COPEC 2010 | HOPE 2012 | |
| isoprene | $(1.8 \pm 1.8) \times 10^9$ | $(2.2 \pm 2.0) \times 10^9$ | $2.7 \times 10^{-11}$ exp(390/T) |
| α-pinene | $(2.7 \pm 3) \times 10^9$ | $(1.5 \pm 1.5) \times 10^9$ | $1.2 \times 10^{-11}$ exp(440/T) |
| β-pinene | $(1.9 \pm 6.6) \times 10^8$ | $(9 \pm 9) \times 10^8$ | $7.4 \times 10^{-11, \text{a,b}}$ |
| 3-carene | $(1.7 \pm 2) \times 10^9$ | $(5.6 \pm 4.7) \times 10^8$ | $8.8 \times 10^{-11, \text{a,b}}$ |
| myrcene | $(2.6 \pm 2.7) \times 10^8$ | $(2.2 \pm 1.6) \times 10^8$ | $3.3 \times 10^{-10, \text{b,c}}$ |
| limonene | n.a. | $(2.9 \pm 2.1) \times 10^8$ | $3 \times 10^{-11}$ exp(515/T),$^{\text{d}}$ |
| sabinene | n.a. | $(9.2 \pm 9.6) \times 10^8$ | $1.2 \times 10^{-10, \text{a,b}}$ |
| γ-terpinene | n.a. | $(1 \pm 1) \times 10^8$ | $1.7 \times 10^{-10, \text{b}}$ |
| MACR | $(1.0 \pm 0.9) \times 10^{10}$ | $(1.4 \pm 0.9) \times 10^9$ | $8 \times 10^{-12}$ exp(380/T) |
| ethanol | $(3.6 \pm 2.2) \times 10^{10}$ | $(1.8 \pm 1.1) \times 10^{10}$ | $3.2 \times 10^{-12}$ exp(20/T) |
| methanol | $(1.0 \pm 1.4) \times 10^{11}$ | $(9.0 \pm 3.4) \times 10^{10}$ | $9 \times 10^{-13, \text{b}}$ |
| ozone | $(1.1 \pm 0.2) \times 10^{12}$ | $(1.1 \pm 0.3) \times 10^{12}$ | $1.7 \times 10^{-12}$ exp(-940/T) |
| SO$_2$ | $(1.4 \pm 1.7) \times 10^{10}$ | $(2.3 \pm 2.2) \times 10^9$ | $2 \times 10^{-12, \text{b}}$ |
| H$_2$O$_2$ | $(1.1 \pm 1.0) \times 10^{10}$ | n.a. | $1.7 \times 10^{-12, \text{b}}$ |
| HO$_2$ | $(9.0 \pm 9.5) \times 10^8$ | $(1.4 \pm 8.6) \times 10^8$ | $4.8 \times 10^{-11}$ exp(250/T) |
| NO | $(6.5 \pm 7.0) \times 10^8$ | $(3.8 \pm 5.0) \times 10^9$ | $1.3 \times 10^{-11, \text{b}}$ |
| NO$_2$ | $(9.5 \pm 5.0) \times 10^9$ | $(3.8 \pm 2.4) \times 10^{10}$ | $1.1 \times 10^{-11, \text{b}}$ |
| CO | $(3.0 \pm 1.2) \times 10^{12}$ | $(2.8 \pm 0.4) \times 10^{12}$ | $2.1 \times 10^{-13, \text{b}}$ |
| HONO | $(3.4 \pm 3.1) \times 10^9$ | n.a. | $6.0 \times 10^{-12, \text{b}}$ |
| propanal | n.a. | $(5.8 \pm 3.0) \times 10^9$ | $4.9 \times 10^{-12}$ exp(405/T) |

| | | | |
|---|---|---|---|
| **acetaldehyde** | $(1.8 \pm 1.0) \times 10^{10}$ | $(2.9 \pm 1.4) \times 10^{10}$ | $1.5 \times 10^{-11,\, b}$ |
| **formaldehyde** | $(1.4 \pm 1.6) \times 10^{10}$ | $(2.1 \pm 0.4) \times 10^{10}$ | $8.5 \times 10^{-12,\, b}$ |
| **acetone** | $(8.2 \pm 3.8) \times 10^{10}$ | $(6.0 \pm 2.2) \times 10^{10}$ | $1.8 \times 10^{-13,\, b}$ |
| **$CH_4$** | $(4.4 \pm 0.07) \times 10^{13}$ | $(4.3 \pm 0.1) \times 10^{13}$ | $6.4 \times 10^{-15,\, b}$ |
| **2-methylpropene** | n.a. | $(4.2 \pm 2.5) \times 10^{8}$ | $6.1 \times 10^{-11,\, a,b}$ |
| **but-1-ene** | n.a. | $(1.4 \pm 4.2) \times 10^{8}$ | $3.1 \times 10^{-11,\, a,b}$ |
| **propene** | n.a. | $(4.7 \pm 3.7) \times 10^{8}$ | $2.9 \times 10^{-11,\, b}$ |
| **cis-2-butene** | n.a. | $(6.1 \pm 3.0) \times 10^{7}$ | $6.4 \times 10^{-11,\, b}$ |
| **ethene** | n.a. | $(7.3 \pm 9.0) \times 10^{9}$ | $7.8 \times 10^{-12,\, b}$ |
| **p-xylene** | n.a. | $(7.2 \pm 5.2) \times 10^{8}$ | $2.0 \times 10^{-11,\, a,b}$ |
| **benzene** | $(2.1 \pm 1.9) \times 10^{9}$ | $(8.0 \pm 4.0) \times 10^{8}$ | $1.2 \times 10^{-12,\, a,b}$ |
| **ethylbenzene** | n.a. | $(2.3 \pm 2.1) \times 10^{8}$ | $7.0 \times 10^{-12,\, a,b}$ |
| **Toluene** | $(6.1 \pm 3.0) \times 10^{9}$ | $(1.2 \pm 0.7) \times 10^{9}$ | $5.6 \times 10^{-12,\, a,b}$ |
| **ethane** | n.a. | $(1.8 \pm 0.3) \times 10^{10}$ | $4.8 \times 10^{-11} \exp(250/T),\ ^{a}$ |
| **propane** | n.a. | $(5.6 \pm 3.6) \times 10^{9}$ | $1.1 \times 10^{-12,\, a,b}$ |
| **methylpropane** | $(1.8 \pm 2.3) \times 10^{9}$ | $(1.4 \pm 0.9) \times 10^{9}$ | $2.1 \times 10^{-12,\, a,b}$ |
| **butane** | $(1.8 \pm 1.6) \times 10^{9}$ | $(2.0 \pm 1.2) \times 10^{9}$ | $2.3 \times 10^{-12,\, a,b}$ |
| **2-methylbutane** | $(1.6 \pm 1.2) \times 10^{9}$ | n.a. | $3.6 \times 10^{-12,\, a,b}$ |
| **n-pentane** | $(1.0 \pm 0.9) \times 10^{9}$ | $(5.6 \pm 5.0) \times 10^{9}$ | $3.8 \times 10^{-12,\, a,b}$ |

a, rate coefficient from (Atkinson and Arey, 2003).
b, at 298 K.
c, rate coefficient from (Hites and Turner, 2009)
d, rate coefficient from (Braure et al., 2014)
1 ppbv = $2.46 \times 10^{10}$ molecules $cm^{-3}$ at 295K and 1013 hPa.

Table SI-3. Average sum of concentrations with 1σ standard deviation of BVOC (isoprene, α- pinene, β-pinene, 3-carene, myrcene, limonene, sabinene, γ-terpinene) and temperature for the entire HOPE 2012 field campaign excluding the period between 26$^{th}$ to 28$^{th}$ of July 2012.

| | $\Sigma$[VOC] [molecules cm$^{-3}$] | Temperature [$^{o}$C] |
|---|---|---|
| HOPE 2012 campaign | $(5 \pm 4) \times 10^9$ | $16 \pm 3$ |
| 26$^{th}$ to 28$^{th}$ of July 2012 | $(1.3 \pm 0.9) \times 10^{10}$ | $22 \pm 3$ |

1 ppbv = 2.46 x 10$^{10}$ molecules cm$^{-3}$ at 295K and 1013 hPa.

[Figure]

[Figure]

[Figure]

[Figure]

Figure SI-1. Time series of trace gases measured during the HOPE 2012 campaign.

[Figure]

Figure SI-2. SCI time series as calculated from the sulfuric acid budget during the HUMPPA-

COPEC 2010 campaign.

[Figure]

Figure SI-3. SCI time series as calculated from the sulfuric acid budget during the HOPE

2012 campaign.

[Figure]

Figure SI-4. SCI time series as calculated from the measured unsaturated VOC  during the

HUMPPA-COPEC 2010 campaign.

[Figure]

Figure SI-5. SCI time series as calculated from the measured unsaturated VOC during the

HOPE 2012 campaign.

[Figure]

Figure SI-6. Contributions of measured trace gases to the measured OH reactivity during the

HUMPPA-COPEC 2010.

[Figure]

Figure SI-7. SCI time series as calculated from the unexplained OH reactivity during the

HUMPPA-COPEC 2010 campaign.

[Figure]

Figure SI-8. Contributions of measured trace gases to the measured OH reactivity during the

HOPE 2012.

[Figure]

Figure SI-9. SCI time series as calculated from the unexplained OH reactivity during the

HOPE 2012 campaign.

[Figure]

Figure SI-10. Background OH as a function of temperature during the HOPE 2012 campaign.

[Figure]

Figure SI-11. Background OH as a function of the ozone concentration during the HUMPPA-

COPEC 2010 campaign.

[Figure]

Figure SI-12. Background OH signal as a function of ozone concentration during the HOPE

2012 campaign.

[Figure]

Figure SI-13. Contribution of measured trace gases to the measured OH reactivity during

HOPE 2012 between the 1st and 3rd of August 2012.

[revised manuscript text omitted]

---

## Author Response (AR2)

We thank the editor for reading the paper carefully and providing thoughtful comments, which have resulted in improvements in the revised version of the manuscript. We reply to each comment below in bold text.

1. Revisit reviewer#1's general comment 1, specifically for the diurnal cycles of OHbg (at least in brief manner).

Following the reviewers suggestions we have included in the current version of the manuscript time series of the SCI obtained from the different estimates. We have now also included in the supplementary information a discussion of these time series, including some information on the diurnal profile. As extensively underlined in the text, the time series carry a large uncertainty due to the many unknowns encountered in their determination. We feel that an analysis of these time series beyond what is currently made available tends towards over-analysis, and certainly has diminished returns with respect to the focus of the paper.

2. Include authors' response to the reviewer#2's 3rd comment with more details than current version of the manuscript.

The authors' response to the 3rd comment from reviewer number 2 was added to the manuscript in more details (Section 4.5).

3. In section 3.3. editor suggests to use instantaneous O3 concentration and redo the calculation than just using one number (7x10-17molec/cm3) for the OH reactivity analysis.

In section 3.3, the instantaneous  $VOC_{unknown}$  and  $O_3$  concentrations are used. We have rephrased the text to make this clearer.

4. For HOPE2012 missing OH reactivity and OHbg analysis, this editor wonders about what if the authors separate the analysis into several periods (i.e. tree cutting, days and nights, 26-08th July, etc.) instead of full mission period. For me it is not convincing to say that the some portion of OHbg is from SCI since two campaign report similar magnitude of SCI concentration but different dependencies of OHbg on temperature and BVOC signature.

In the current version of the manuscript, for the HOPE 2012 campaign, there are some data periods separated from the rest of the campaign (i.e. tree cutting, 26th-28th of July) as they were characterized by peculiar behaviors (e.g. larger OH reactivity, instrument left unattended). We do not have valid reasons to divide the data into even more periods, nor do we feel that this would help give a better idea of the reasons for the discrepancy between the two environments.

Regarding SCI as a source of the  $OH_{bg}$  signal: we state in several places (Page 24, lines 18-19, Page 27, lines 21-22, Page 33, lines 23-24) that the discrepancy in the behavior of the  $OH_{bg}$  between the two

environments could be due to the contribution of more species to the  $OH_{bg}$  during the HOPE 2012 campaign compared to HUMPPA 2010, but likewise we do not believe that the evidence shown allows us to completely exclude a contribution of SCI to the  $OH_{bg}$  for either campaign. We show clearly that the  $OH_{bg}$  signals observed in the two environments cannot be compared from a point of view of absolute value. It is correct that during the night the amount of  $OH_{bg}$  observed in counts per second normalized on laser power in the two environments is similar, but this does not take into account the sensitivity towards the species causing the  $OH_{bg}$ , which could have been higher in the HOPE 2012 campaign compared to the HUMPPA 2010 one. In addition, as mentioned, in the night, during the HOPE 2012 campaign, there could have been additional interference caused by NO3. Finally, our paper also discusses the mechanistic discrepancy that arises from assigning an absolute OH concentration to the  $OH_{bg}$  signal (section 4.6), as this would imply a massive OH source in the atmosphere that is unsupported by any data.

Combined, we feel that the paper is very open about the fact that the  $OH_{bg}$  may not be fully caused by SCI, and that the SCI concentration derived by  $OH_{bg}$  is merely indicative for the hypothetical case where SCI are the sole interference. To date, no other interferences other than  $NO_3$  have been unambiguously identified, such that this hypothesis remains plausible.

\*Technical corrections

1. Double check significant figures

Checked

2. p19 line 5, OH reactivity ==> OH production

**Corrected**

**Estimating the atmospheric concentration of Criegee intermediates and their possible interference in a FAGE-LIF**

**3 instrument**

Anna Novelli1,2, Korbinian Hens1, Cheryl Tatum Ernest1,3, Monica Martinez1, Anke C. 4 Nölscher1,4, Vinayak Sinha5, Pauli Paasonen6, Tuukka Petäjä6, Mikko Sipilä6, Thomas Elste7, 5 Christian Plass-Dülmer7, Gavin J. Phillips1,8, Dagmar Kubistin1,7,9, Jonathan Williams1, Luc 6 Vereecken1,2, Jos Lelieveld1 and Hartwig Harder1 7 8 9 [1] {
[revised manuscript text omitted]

| $H_2SO_4^{a}$                       | $(2.0 \pm 2.0) \ge 10^6$    | $(8.5 \pm 8.5) \ge 10^5$    |
| $OH^{a}$                            | $(7.0 \pm 8.0) \ge 10^5$    | $(1.6 \pm 1.6) \ge 10^6$    |
| $O_3^{a}$                           | $(1.1 \pm 0.2) \ge 10^{12}$ | $(1.1 \pm 0.3) \ge 10^{12}$ |
| $\Sigma[\text{VOC}]^{a,b}$          | $(7.3 \pm 7.1) \ge 10^9$    | $(9.8 \pm 9.0) \ge 10^9$    |
| OH Reactivity c          | $9.0 \pm 7.6$               | $3.5 \pm 3.0$               |
| Condensation sink (CS) c | $(10 \pm 4.0) \ge 10^{-3}$  | $(7.0 \pm 3.0) \ge 10^{-3}$ |

a, Units: molecules  $cm^{-3}$ .

b, HUMPPA COPEC 2010: isoprene,  $(-)/(+) \alpha$ -pinene,  $(-)/(+) \beta$ -pinene, 3-carene, and myrcene.

HOPE 2012: isoprene,  $\alpha$ -pinene,  $\beta$ -pinene, 3-carene, myrcene, limonene, 2-

methylpropene, but-1-ene, sabinene,  $\gamma$ -terpinene, propene, cis-2-butene and ethene. c, Units: s-1.

 $1 \text{ ppbv} = 2.5 \text{ x } 10^{10} \text{ molecules cm}^{-3} \text{ at } 295 \text{K} \text{ and } 1013 \text{ hPa.}$

- 1 Table 2. SCI estimates for the HUMPPA-COPEC 2010 and HOPE 2012 campaigns. Average
- 2 concentration (molecule cm-3), with  $1\sigma$  variability.

| Approach                                                                                                                                                                                                                                                                                                                                                                                                       | HUMPPA-COPEC
2010                                | HOPE 2012                    |
|----------------------------------------------------------------------------------------------------------------------------------------------------------------------------------------------------------------------------------------------------------------------------------------------------------------------------------------------------------------------------------------------------------------|-----------------------------------------------------|------------------------------|
| М. : Н со                                                                                                                                                                                                                                                                                                                                                                                                      | $(2.3 \pm 2.0) \ge 10^4 a$                          | $(2.0 \pm 3.0) \ge 10^4 a$   |
| Missing $H_2SO_4$                                                                                                                                                                                                                                                                                                                                                                                              | $(1.6 \pm 2.0) \ge 10^{6}$ b                        | $(1.0 \pm 3.0) \ge 10^{6}$ b |
| Measured unsaturated VOC                                                                                                                                                                                                                                                                                                                                                                                       | $(5.0 \pm 4.0) \ge 10^3$                            | $(7.0 \pm 6.0) \ge 10^3$     |
| Unexplained OH reactivity                                                                                                                                                                                                                                                                                                                                                                                      | $(1.0 \pm 1.0) \ge 10^5$                            | $(2.0 \pm 1.5) \ge 10^4$     |
|                                                                                                                                                                                                                                                                                                                                                                                                                | $(2.0 \pm 2.0) \ge 10^4$ c                          | n. a.                        |
| Unexplained OH production                                                                                                                                                                                                                                                                                                                                                                                      | $(4.0 \pm 4.0) \ge 10^{5}$ d                        | n. a.                        |
| a, $k_{SCI+SO2} = 3.3 \times 10^{-11} \text{ cm}^3 \text{ molecule}^{-1} \text{ s}^3$
b, $k_{SCI+SO2} = 5.0 \times 10^{-13} \text{ cm}^3 \text{ molecule}^{-1} \text{ s}^3$
c, $P_{OH}^{un \exp lained} = 1.0 \times 10^6$ molecule cm -3 s
d, $P_{OH}^{un \exp lained} = 2.0 \times 10^7$ molecule cm -3 s
1 ppbv = 2.5 x 10 10 molecules cm -3 at 29 | -1
3 1
5K and 1013 hPa. |                              |